# Structural basis of catalytic activation in human splicing

Jana Schmitzová[1,6,8], Constantin Cretu[1,2,7,8], Christian Dienemann[3], Henning Urlaub[4,5] & Vladimir Pena[1,2 ✉]

Pre-mRNA splicing follows a pathway driven by ATP-dependent RNA helicases. A crucial event of the splicing pathway is the catalytic activation, which takes place at the transition between the activated B[act] and the branching-competent B[*] spliceosomes. Catalytic activation occurs through an ATP-dependent remodelling mediated by the helicase PRP2 (also known as DHX16)[1–3]. However, because PRP2 is observed only at the periphery of spliceosomes[3–5], its function has remained elusive. Here we show that catalytic activation occurs in two ATP-dependent stages driven by two helicases: PRP2 and Aquarius. The role of Aquarius in splicing has been enigmatic[6,7]. Here the inactivation of Aquarius leads to the stalling of a spliceosome intermediate—the B[AQR] complex—found halfway through the catalytic activation process. The cryogenic electron microscopy structure of B[AQR] reveals how PRP2 and Aquarius remodel B[act] and B[AQR], respectively. Notably, PRP2 translocates along the intron while it strips away the RES complex, opens the SF3B1 clamp and unfastens the branch helix. Translocation terminates six nucleotides downstream of the branch site through an assembly of PPIL4, SKIP and the amino-terminal domain of PRP2. Finally, Aquarius enables the dissociation of PRP2, plus the SF3A and SF3B complexes, which promotes the relocation of the branch duplex for catalysis. This work elucidates catalytic activation in human splicing, reveals how a DEAH helicase operates and provides a paradigm for how helicases can coordinate their activities.

Pre-mRNA splicing occurs in two transesterification reactions on the spliceosome, the sequential assembly and remodelling of which are driven by eight conserved ATP-dependent RNA helicases[4,8–10]. The branching reaction occurs between the branch site adenosine (BS-A) and the 5' splice site (5'SS) during the formation of the post-branching complex (also known as the C complex). As a prerequisite for branching, the reactants juxtapose to the catalytic centre in a process known as catalytic activation, which occurs at the conversion of the B[act] (also known as activated spliceosome) to the branching-competent B[*] (also known as catalytically activated spliceosome) complex[1]. The B[*] and the post-branching C complexes are very similar in structure and composition, differing primarily in the phosphodiester bond between the BS-A and the 5'SS[10].

Catalytic activation requires the remodelling of B[act] spliceosomes. As part of the U2–BS duplex (also known as the branch duplex), the BS-A is sequestered 50 Å away from the catalytic centre and is enclosed by the SF3B complex of the U2 small nuclear ribonucleoprotein (snRNP)[2,3]. This inactive conformation of B[act] is stabilized by the RES complex and the proteins SF3B1 and PRP8. Additionally, the proteins SF3A2 and CWC24 (also known as RNF113A) shield the 5'SS from the catalytic centre. These contacts must be disrupted during catalytic activation to relocate the branch helix to the active centre for catalysis[2,11–13].

In *Saccharomyces cerevisiae* (budding yeast), the DEAH helicase Prp2p drives catalytic activation by releasing RES, SF3a and SF3b complexes, and the proteins Cwc24p and Cwc27p; however, mechanistic details of this process remain largely unknown[1,5,14]. In humans, in addition to PRP2 (an orthologue of Prp2p), the helicase Aquarius is recruited to the spliceosome as part of a pentameric intron-binding complex (IBC), which contributes to the B-to-C transitions of the spliceosome. The spliceosome remodelling promoted by Aquarius and the time point of its action relative to PRP2 is unknown[6,7]. Aquarius also stands out as the only splicing helicase from the SF1 family.

Here we reconstitute a spliceosome stalled between the actions of PRP2 and Aquarius, which we refer to as the B[AQR] complex. The cryogenic electron microscopy (cryo-EM) structure of the B[AQR] spliceosome reveals the mechanism of B[act]-to-B remodelling and the mode of action of PRP2 assisted by interacting proteins. Our results also provide insight into how the two helicases coordinate their activities to promote the branching reaction.

## Overview of the human B[AQR] spliceosome
After identifying the IBC as a complex that delivers Aquarius to the spliceosome[6], we reconstituted the B[AQR] spliceosome on MINX

[1]Macromolecular Crystallography, Max Planck Institute for Multidisciplinary Sciences, Göttingen, Germany. [2]Research Group Mechanisms and Regulation of Splicing, The Institute of Cancer Research, London, UK. [3]Molecular Biology, Max Planck Institute for Multidisciplinary Sciences, Göttingen, Germany. [4]Bioanalytical Mass Spectrometry, Max Planck Institute for Multidisciplinary Sciences, Göttingen, Germany. [5]Institute of Clinical Chemistry, Bioanalytics, University Medical Center Sciences, Göttingen, Germany. [6]Present address: Molecular Biology, Max Planck Institute for Multidisciplinary Sciences, Göttingen, Germany. [7]Present address: Cluster of Excellence Multiscale Bioimaging (MBExC), Universitätsmedizin Göttingen, Göttingen, Germany. [8]These authors contributed equally: Jana Schmitzová, Constantin Cretu. ✉e-mail: Vlad.Pena@icr.ac.uk

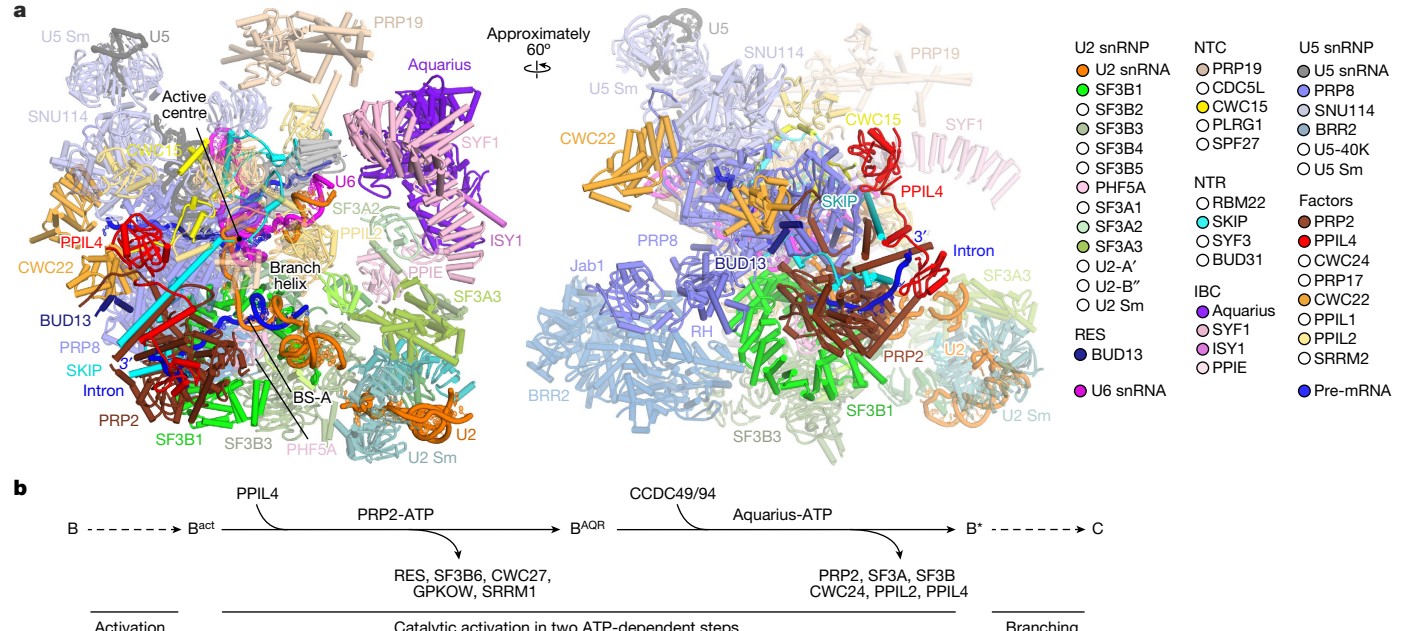

**Fig. 1 | Structural organization of the B<sup>AQR</sup> spliceosome. a**, Overview of the B<sup>AQR</sup> cryo-EM structure. Key subunits are colour coded. **b**, Compositional remodelling of the human spliceosome during catalytic activation. Subunits recruited or destabilized by the ATPase activities of PRP2 and Aquarius are indicated. Destabilized subunits can remain flexibly attached to the spliceosomes, often at lower stoichiometry. NTR, NTC-related proteins.

pre-mRNA in HeLa nuclear extracts supplemented with IBC carrying the dominant-negative Aquarius mutant K829A (IBC(K829A)). This mutant stalls the spliceosome before branching[6] (Extended Data Fig. 1a–d and Supplementary Fig. 1). The proteome of B<sup>AQR</sup> resembled one of the B<sup>act</sup> complexes[2,13,15]. However, within B<sup>AQR</sup>, PRP2 and PPIL4 were abundant, whereas the PRP2 cofactor GPKOW was detected in low amounts (Supplementary Data 1).

We reconstructed a cryo-EM map of the B<sup>AQR</sup> core complex at about 2.9 Å resolution. Focused 3D classification allowed us to resolve the peripheral regions of the complex and model in the BRR2 helicase, the PRP19 and IBC complexes, and the 3′ module of the U2 snRNP (Fig. 1a, Extended Data Figs. 1e–i, 2 and 3a–d, Extended Data Table 1, Supplementary Data 2 and Supplementary Video 1).

Compared with B<sup>act</sup> complexes[2,13], B<sup>AQR</sup> exhibited repositioning of PRP2, the SF3B complex, SKIP (also known as PRP45), the Jab1 domain and the RNaseH-like (RH) domain of PRP8, and the intron RNA downstream of the branch helix. More subtle changes were present for BRR2, CWC15 (also known as AD002) and CWC22. Furthermore, PPIL4 was recruited to the B<sup>AQR</sup> complex while the RES complex, SF3B6 (also known as p14), CWC27, SRRM1 and GPKOW were destabilized (Fig. 1 and Supplementary Video 1).

The position of Aquarius was primarily the same in B<sup>act</sup> complexes and B<sup>AQR</sup> complexes[2,13], but notably different in C complexes[2,16], which indicates that inactivated Aquarius prevents the conversion of B<sup>AQR</sup>-to-C complexes (Extended Data Fig. 3e). By contrast, PRP2 occupied different locations in B<sup>act</sup> and B<sup>AQR</sup>. This result indicates that B<sup>AQR</sup> is an intermediate spliceosome stalled after the translocation of PRP2 but before the action of Aquarius. We conclude that B<sup>AQR</sup> represents an intermediate of the splicing pathway that follows the B<sup>act</sup> complex while preceding B* and C complexes[1,11,12,17] (Fig. 1b).

## Structure of PRP2 in the B<sup>AQR</sup> spliceosome

PRP2 is composed of a flexible N-terminal region (residues 1–387) and a conserved globular core (PRP2<sup>core</sup>, residues 388–1042), which is typical of all four spliceosomal DEAH helicases (Fig. 2a). Structures of fungal PRP2<sup>core</sup> have been reported for *Chaetomium thermophilum*

and budding yeast in complex with ligands such as ADP, ADP-BeF<sub>3</sub><sup>−</sup>, RNA oligonucleotides and the cofactor Spp2p (refs. 5,18). The PRP2<sup>core</sup> has also been resolved at 3.2 Å resolution in yeast B<sup>act</sup> spliceosomes or docked as a homology model in human B<sup>act</sup> complexes[5,11,13].

The cryo-EM density map of human B<sup>AQR</sup> enabled the building of the PRP2<sup>core</sup> and of the accessory N-terminal domain (PRP2<sup>NTD</sup>, residues 265–295; Fig. 2b,c and Extended Data Fig. 3c,d). The PRP2<sup>core</sup> region exhibited the two RecA-like domains (RecA1 and RecA2) and a carboxy-terminal module (PRP2<sup>CTD</sup>) that encompassed the winged-helix (WH), oligonucleotide/oligosaccharide-binding (OB) and helix-bundle (HB) domains. The PRP2<sup>core</sup> from B<sup>AQR</sup> is in the open, post-ATP hydrolytic conformation. This conformation has previously only been observed for the ATPase-defective mutant Prp2p(K252A) in the yeast B<sup>act</sup> spliceosome[5].

The structure and function of PRP2<sup>NTD</sup> were unknown. In the B<sup>AQR</sup> spliceosome, PRP2<sup>NTD</sup> acquired a stage-defining fold that lacked a hydrophobic core and was organized into three distinct modules. Owing to their extended conformation, they interacted with several spliceosomal subunits. We refer to these elements as the hook, the clip and the pin (Fig. 2a,b and Extended Data Fig. 4a).

## PRP2 translocates about 19 nucleotides

PRP2 has previously been observed at the periphery of human and yeast B<sup>act</sup> complexes, residing on the convex side of the HEAT domain of SF3B1 (SF3B1<sup>HEAT</sup>)[5,11,13]. Notably, in the B<sup>AQR</sup>, PRP2 is no longer anchored at the periphery of the spliceosome. Instead, PRP2 has now moved around 85 Å deep towards the branch helix and the core of B<sup>AQR</sup>, filling a large cavity between SF3B1 and PRP8 within B<sup>AQR</sup> (Fig. 2d). In the new location, PRP2 shares interfaces with SF3B1<sup>HEAT</sup> and PRP8 through the RecA2 and HB domains, respectively (Extended Data Fig. 4b,c). The helicase rotated about 70° counterclockwise, and the RecA-like domains adopted the open conformation on top of SF3B1<sup>HEAT</sup> (refs. 5,18) (Fig. 2b,c). In this post-translocation state, PRP2 accommodated seven nucleotides of the intron, assigned as the 7–13 region downstream of BS-A, in a channel framed by the RecA-like domains on one side and PRP2<sup>CTD</sup> on the other side.

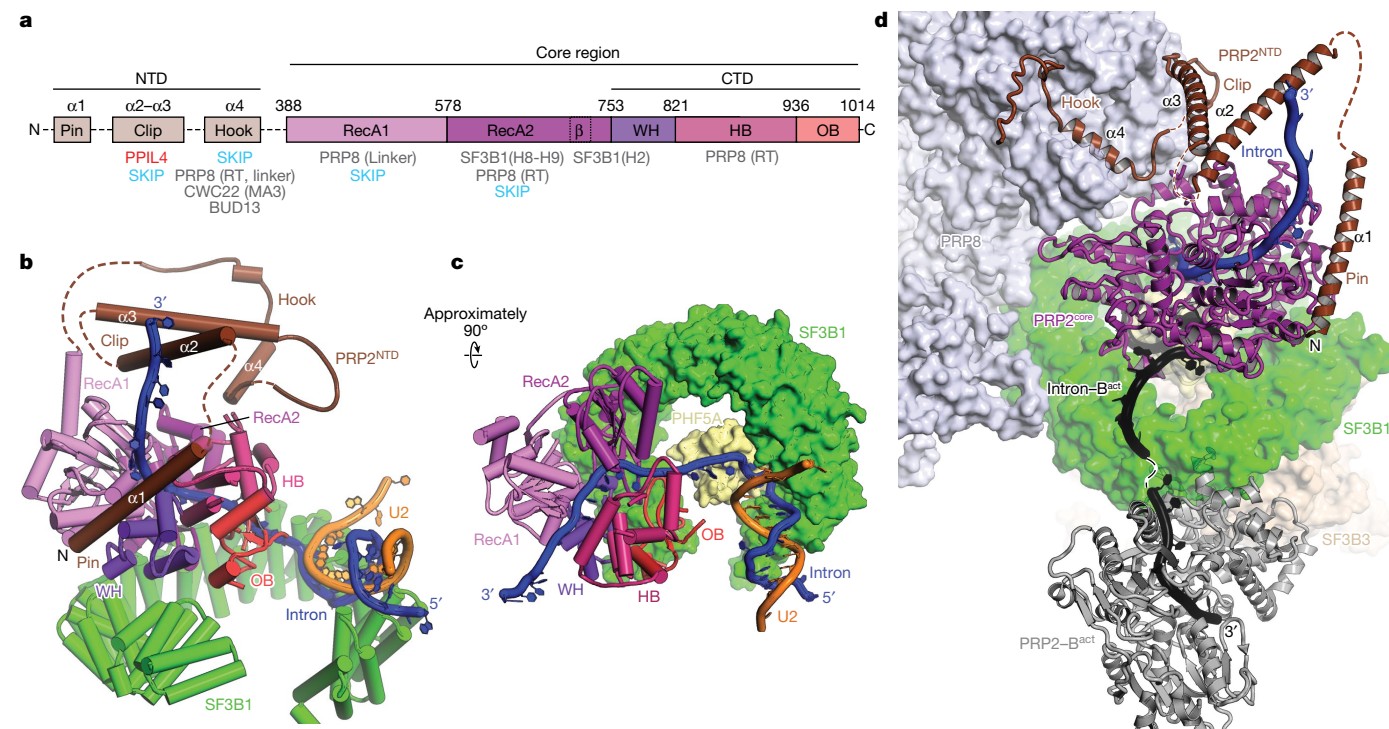

**Fig. 2 | Structure and translocation of PRP2 during catalytic activation.**
**a**, Domain composition and interactions of PRP2 in the B$^{AQR}$ complex.
**b,c**, Structure and conformation of PRP2, the intron and SF3B1 in B$^{AQR}$. Note that PRP2$^{NTD}$ (the N-terminal domain) is not visible in **c**. **d**, In B$^{AQR}$, PRP2 is moved in a cavity framed by PRP8 and SF3B1. The previous location and conformation of PRP2 and the intron in B$^{act}$ are shown in grey and black, respectively.

Within yeast B$^{act}$ complexes, PRP2 binds RNA at positions 28–34 (refs. 5,19), which probably correspond to positions 26–32 in human B$^{act}$ complexes according to our 3D alignment data (Extended Data Fig. 5a–d). A comparison between B$^{AQR}$ and B$^{act}$ indicated that PRP2 advanced approximately 19 nucleotides towards the spliceosome core during the B$^{act}$-to-B$^{AQR}$ transition, which explains the stringent requirement for at least 32–34 nucleotides downstream of BS-A for efficient branching in human splicing[15] (Figs. 2d and 3a and Extended Data Fig. 5e). Therefore, PRP2 translocates along the intron from its earlier position in the B$^{act}$ complex rather than acting from a distance like a molecular winch that pulls the intron from the periphery of the spliceosome (Supplementary Video 2). Concomitantly, the exiting intron was relocated about 70 Å owing to the helicase rotation and exchange in intron–protein interactions following the formation of the B$^{AQR}$ complex (Fig. 2d and Extended Data Fig. 5f).

## Mechanism of dissociation of the RES complex

The RES complex is required to convert B to B$^{act}$ complexes and dissociates after PRP2 action at the B$^{act}$-to-B* transition[20,21]. The RES complex comprises the proteins BUD13, RBMX2 (also known as SNU17) and SNIP1 (also known as PML1), and resides in a cavity of the B$^{act}$ spliceosome framed by SF3B1 and PRP8. In this cavity, the RES complex binds the proteins SF3B1, PRP8, SKIP and CWC22, whereas the RRM domain of RBMX2 binds the polypyrimidine tract (PPT; or the equivalent region in yeast)[2,13].

In the B$^{AQR}$ structure, the translocation of PRP2 displaces several subunits, including the RES complex from the top of SF3B1$^{HEAT}$ (Fig. 3b,c and Supplementary Video 3). The dissociation of RES is primarily caused by the stripping of RBMX2 away from the intron (Fig. 3a and Extended Data Fig. 5e,f). As BUD13 and SNIP1 share interfaces with RBMX2, dislocation of the latter probably releases the entire RES complex. Moreover, the clip of PRP2$^{NTD}$ replaced SNIP1 by interacting with the α3 helix of SKIP, whereas new interactions between PRP2$^{core}$, SKIP and PRP8 replace those between BUD13 and SKIP in the B$^{act}$ complex (Fig. 3a and Extended Data Fig. 4c,d).

After destabilizing the RES complex from RNA and former protein contacts, only a short stretch of BUD13 (residues 530–557) is visible in B$^{AQR}$, which is bound to a composite structure formed by the hook of PRP2$^{NTD}$, PRP8 and the slightly relocated MA3 domain of CWC22 (Fig. 3a and Extended Data Fig. 4e,f). However, proteomics analysis showed that the entire RES complex remained bound to the spliceosome (Supplementary Data 1). Thus, BUD13(530–557) anchors the destabilized RES complex to the spliceosome, after being displaced from the intron. This flexible anchoring is conditioned by the hook of PRP2 binding to PRP8, the presence of which may serve as a signature for the occurrence of the translocation of PRP2. In this way, PRP2 might control the trajectory of the exiting RES complex, moving it away from the downstream intron region that must engage in new interactions.

## SF3B1 unfastens the branch duplex

The remodelling and destabilization of the SF3B complex is a key event in the splicing pathway and a consequence of the translocation of PRP2. The largest subunit of SF3B is SF3B1, of which the HEAT domain exhibits a distinctive loose conformation in the B$^{AQR}$ complex. This conformation differs markedly from the open and closed conformations of SF3B1 from pre-A and from A-to-B$^{act}$ complexes, respectively (Fig. 3d). For example, SF3B1(A514) and MINX(U136) are separated by 4, 48 and 42 Å in B$^{act}$, B$^{AQR}$ and pre-A complexes, respectively. Furthermore, interacting residues SF3B1(H550) and PHF5A(Y51) from pre-A complexes are separated by 16 Å in the B$^{AQR}$ complex.

The transition of SF3B1 from closed to loose following the conversion of B$^{act}$ to B$^{AQR}$ seemed to be caused by PRP2, after stripping the PPT (nucleotides 13–18) away from the binding pocket. Because the pocket is also a hinge, SF3B1 opens widely to unclamp the branch duplex from

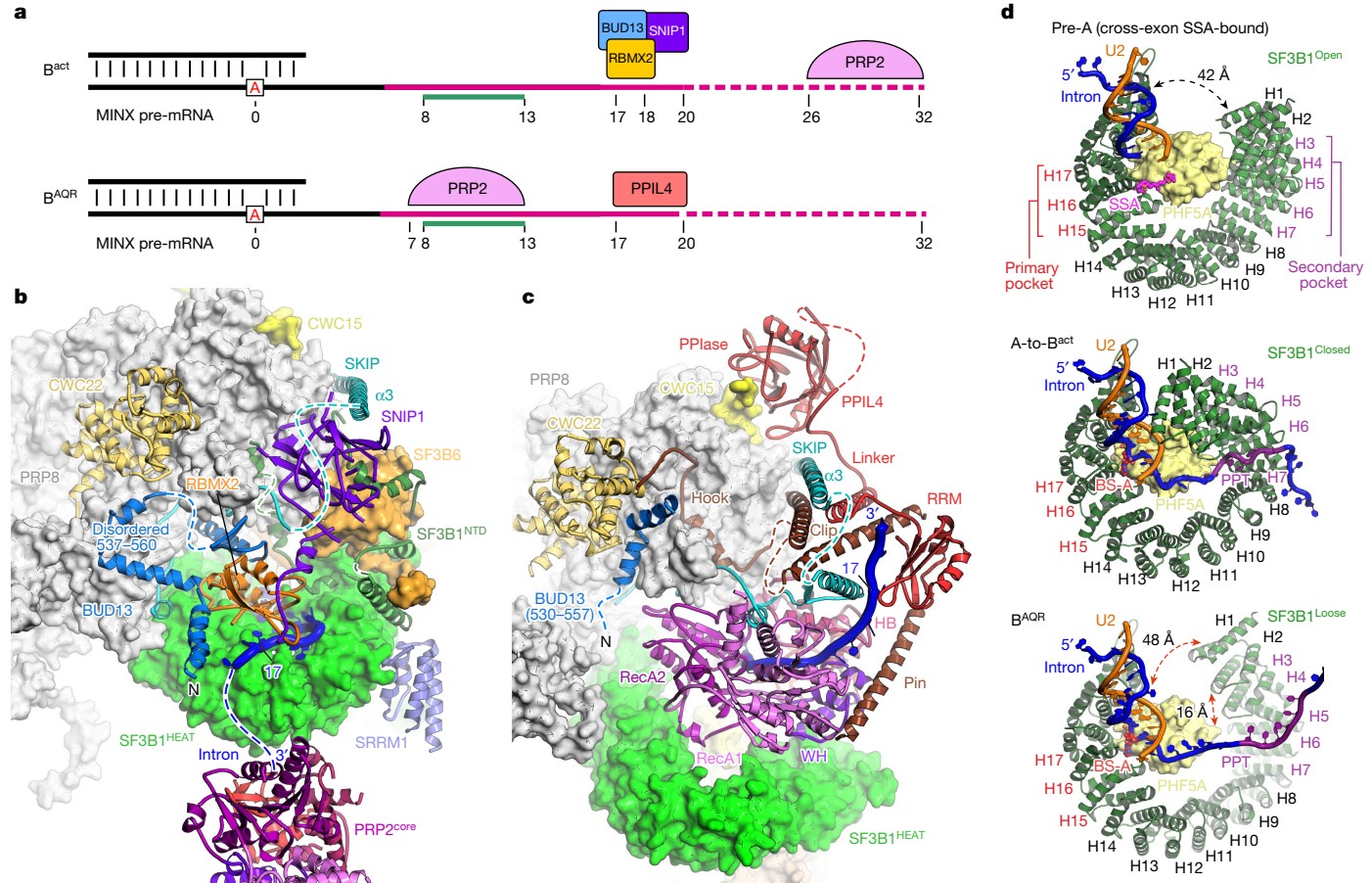

**Fig. 3 | PRP2 tightly regulates the dissociation of the RES complex and remodels the SF3B complex. a**, Relocation of PRP2 on the intron, RES complex dissociation from the translocated RNA and subsequent binding of PPIL4 to the latter. Pink, green and broken lines represent the PPT, the PPT region bound by SF3B1 within B$^{act}$, and the PPT-equivalent region visible only within the budding yeast B$^{act}$ structure, respectively. **b**, Close-up view of the interfaces between PRP2 and other subunits in the B$^{act}$ complex. **c**, The equivalent view and orientation of the B$^{AQR}$ complex. **d**, SF3B1 exhibits primary and secondary hinged pockets for binding of BS-A and PPT, respectively. SF3B1 adopts a loose conformation in B$^{AQR}$, whereby the primary and secondary pockets are occupied and free, respectively.

one side, like an opening pair of tweezers (Figs. 2b,c and 3d, Extended Data Fig. 6a,b and Supplementary Video 3).

The transition mainly involvs the H1–H10 repeats, whereas the contacts between BS-A and its binding pocket are almost identical in B$^{act}$ and B$^{AQR}$ complexes (Fig. 3d and Extended Data Fig. 6c). Overall, the consequence of the translocation of PRP2 is the disruption of 60% (around 1,100 Å$^2$) SF3B1's contact interface with RNA, facilitating the release of the branch duplex for relocation to the catalytic centre.

The opening of SF3B1$^{HEAT}$ by PRP2 triggers additional remodelling that weakened the interaction between the SF3B complex and the domains of PRP8 (Supplementary Video 3). In detail, the change in SF3B1$^{HEAT}$ curvature disrupts contacts between the repeats H10–H11 and PRP8$^{RH}$, and perturbs the endonuclease-like domain of PRP8 (Extended Data Fig. 6a). In addition, transition to the loose conformation induces the release of SRRM1 and SF3B6 from SF3B1. De-structuring of SF3B1$^{NTD}$ (residues 1–488) also occurrs, which is an extended region that wraps around SF3B6 within the B$^{act}$ complex (Fig. 3b,c and Supplementary Video 3). As SF3B6 and SF3B1$^{NTD}$ are connectors between the SF3B complex and PRP8 (Fig. 3c), their destabilization probably facilitates the subsequent release of SF3B and SF3A complexes, and the Aquarius-dependent relocation of the branch duplex at the transition from B$^{AQR}$ to B$^*$ and then C complexes (Supplementary Video 4).

## Termination of the translocation of PRP2

The low complementarity between U2 and the branch sites of human introns requires tight regulation of the activity of PRP2 to prevent it from unwinding the branch duplex. Therefore, PRP2 needs to advance sufficiently far to dissociate the PPT from SF3B1 and open the SF3B1$^{HEAT}$ clamp, yet not too far to compromise the branch duplex. Although PRP8 acts like an initial physical barrier able to stop the movement of PRP2 (Fig. 2d), the helicase may keep pulling the intron like a winch to unwind or alter the branch duplex. Thus, a different mechanism for the termination of PRP2 translocation is required.

PRP2 in B$^{AQR}$ shares large interfaces with PPIL4 (1,040 Å$^2$) and SKIP (1,850 Å$^2$), which suggests that these proteins could regulate the helicase (Fig. 3c and Extended Data Fig. 4g–k). Indeed, flexible elements of PRP2$^{NTD}$, PPIL4 and SKIP form a stable 3D assembly that bound the exiting RNA strand at the nucleotides 17–20, downstream of the BS-A (Fig. 4a and Extended Data Fig. 4g,h). This result indicates that these proteins have a collective role in the timely termination of helicase translocation.

PPIL4 belongs to the cyclophilin family of peptidylprolyl isomerases (PPIases) and was not assigned in other cryo-EM structures of spliceosomes, which left its role in splicing unknown. In the B$^{AQR}$ map, we built PPIL4, which included the PPIase and RRM domain separated

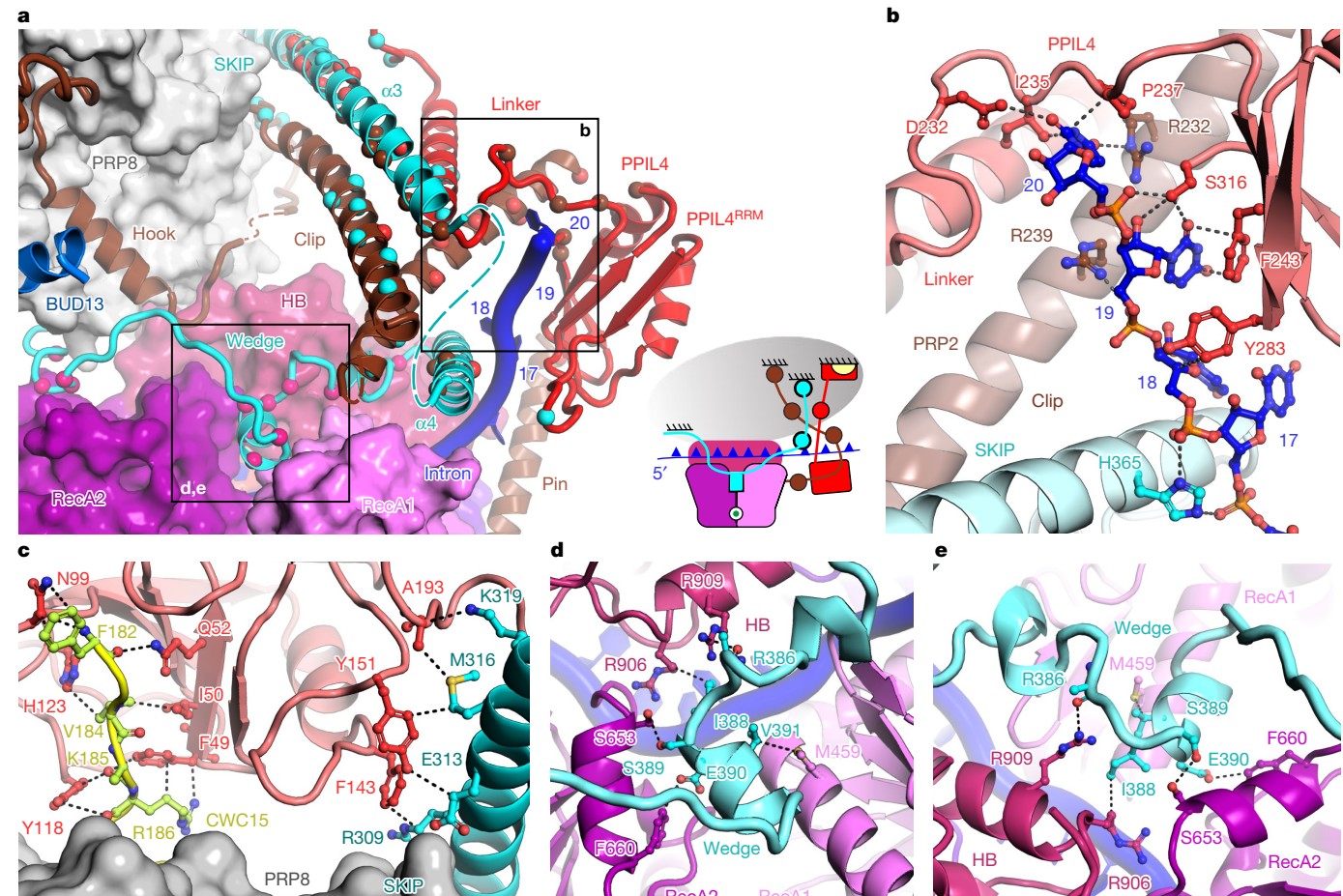

**Fig. 4 | PRP2 translocation terminates in a 3D construction reminiscent of a molecular brake. a**, Interactions between the subunits of the molecular brake. Interacting residues are shown as spheres depicted in the same colour as their binding partner. The domains of PRP2 RecA1, RecA2, HB and NTD are depicted. The anchoring surfaces of the brake elements to PRP8 (grey oval) are indicated. The inset shows a schematic representation of the molecular brake, in which the subunits are coloured as in the surface and cartoon representation. CWC15 is yellow. **b**, Interactions of the intron with PRP2$^{NTD}$, PPIL4 and SKIP. **c**, Interactions of PPIL4 with CWC15 and SKIP. **d,e**, The wedge element of SKIP intercalates between RecA1, RecA2 and the HB domains of PRP2, indicating inhibition of translocation. PPIL4, SKIP and CWC15 are red, cyan and yellow, respectively.

by a linker containing an α-helix (Extended Data Fig. 4g,h). Notably, PPIL4$^{RRM}$ and the linker bound the exiting intron, covering the position occupied previously by RBMX2$^{RRM}$ of the RES complex in B$^{act}$ (Figs. 3a–c and 4a,b). Therefore, this intron-binding site is only accessible for PPIL4 after PRP2 has dissociated RES and translocated to an upstream position. Consistent with the B$^{AQR}$ structure, recombinant PPIL4 formed a stable complex with PRP2 (137–1022) (delineated as the region built in the cryo-EM map; Extended Data Fig. 7a–c and Supplementary Fig. 2). Furthermore, PPIL4 was able to bind a model RNA substrate (for example, a poly(A)-rich 3′ single-stranded overhang) both in the presence and absence of PRP2 (Extended Data Fig. 7d,e and Supplementary Fig. 3).

Anchoring of PPIL4 to spliceosomes is facilitated by the NineTeen complex (NTC). The CWC15 component of NTC resides on the surface of PRP8, where it binds the PPIL4$^{PPIase}$ domain through a short stretch (Figs. 3c and 4c and Extended Data Fig. 4g). The defined distance between the PPIase and RRM domains of PPIL4 is maintained by the interdomains linker, structurally stabilized by helices from PRP2$^{NTD}$ and SKIP (Fig. 4a and Extended Data Fig. 4h).

SKIP is a largely unstructured protein, whose disparate elements interact with subunits of other spliceosomes[8]. The long characteristic α3 helix of SKIP (residues 282–340) binds PRP8 in B$^{act}$, B$^{AQR}$ and C complexes (Fig. 3b,c and Extended Data Fig. 8a). In B$^{AQR}$, this helix stabilizes

the post-translocation state of PRP2 through interactions with the clip of PRP2$^{NTD}$ and the linker of PPIL4. Furthermore, the residues 352–400 of SKIP are visible only in B$^{AQR}$, in which the α4 helix binds the clip of PRP2$^{NTD}$ and the intron at nucleotides 17–20 (Fig. 4a,b and Extended Data Fig. 4h). Notably, the residues 371–400 binds PRP2$^{core}$, such that a 3$_{10}$ helix intercalates like a wedge between the two RecA modules and the HB domain of PRP2 (Fig. 4a and Extended Data Fig. 4j). Superposition of our structure with the PRP2 structure from *C. thermophilum* in the closed state (Protein Data Bank (PDB) identifier 6zm2)[18] indicates how the wedge element of SKIP might lock the helicase in the open conformation and therefore terminates translocation (Fig. 4d,e, Extended Data Fig. 4i and Supplementary Video 5). We suggest that PPIL4 and SKIP might act as negative regulators of PRP2, forming a context-specific molecular brake to terminate the translocation of the helicase at a defined time and location on the intron.

## Mode of action of a DEAH helicase

The four DEAH splicing helicases (PRP2, PRP16, PRP22 and PRP43) were observed at relatively fixed positions from the periphery of spliceosomes, where they act by less understood mechanisms[4]. In particular, whether PRP2 moves along the RNA or pulls the substrate from a distance remained unclear[3,4,19,22,23].

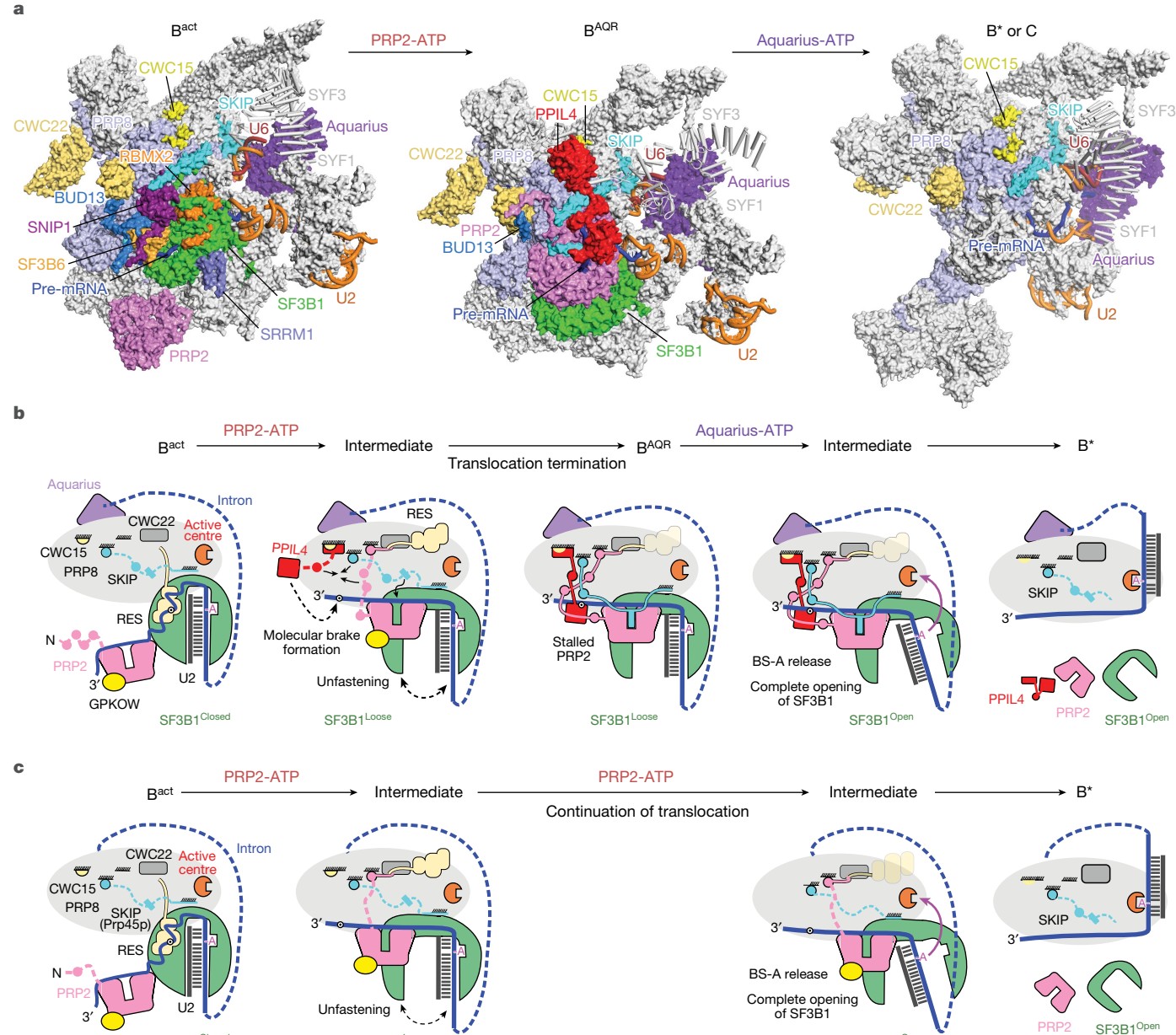

**Fig. 5 | Model of catalytic activation in pre-mRNA splicing. a**, Catalytic activation depicted as structural transitions between B[act], B[AQR] and B*. B* and the post-branching C complex have similar structures. Most subunits are shown as surface representations. SYF1 and SYF3 are shown as ribbons to enable better visualization of Aquarius. **b**, Structure-based model of human spliceosome remodelling by PRP2 and Aquarius. The hypothetical intermediate between B[act] and B[AQR] considers the following: (1) the molecular brake can form only after the binding of PPIL4 to the RNA, the binding site of which becomes available after dissociation of the RES complex by the translocation of PRP2; and (2) the advance of PRP2 should start stripping the intron from SF3B1 and promote the loose conformation of SF3B1. The hypothetical intermediate between B[AQR] and B* was generated by considering that Aquarius should induce complete displacement of the branch duplex from SF3B1, liberating BS-A from its binding pocket. Consequently, SF3B1 will transit from the loose to the open conformation. Because the HEAT repeats that bind PRP8 (in B[AQR]) rearrange after loose-to-open conformation, SF3B1 dissociates from PRP8. This likely causes the complete destabilization of the SF3A–SF3B complexes and PRP2 from the spliceosome. **c**, Model of the spliceosome remodelling by Prp2p in budding yeast *(S. cerevisiae)*, based on the similarities and differences with the human counterpart.

The structure of B[AQR] reveals how PRP2 can operate in several stages (Fig. 5 and Supplementary Videos 2, 3 and 5). Starting from the periphery of B[act] complexes[5,11,13], PRP2 translocates around 19 nucleotides to form the B[AQR] complex while stripping away the proteins RBMX2 and SF3B1 from the intron. A cascade of events follows, including destabilizations (RES proteins), dissociations (SF3B6, SRRM1 and CWC27), restructuring (SF3B1[NTD], SKIP and PRP2[NTD]), rearrangement of globular domains (PRP8 and CWC22) and recruitment (PPIL4). Essentially, the signal from ATP hydrolysis is amplified to induce large-scale remodelling.

The translocation of PRP2 terminates six nucleotides downstream of the BS-A in a network of interactions that resemble a molecular brake constructed from PRP2[NTD], SKIP and PPIL4. The brake binds the exiting intron like a brake shoe and intercalates a wedge element of SKIP between the RecA1, RecA2 and HB domains, immobilizing PRP2 in the open conformation (Fig. 4a,b,d and Supplementary Video 5). Thus, PPIL4 and SKIP might act as helicase cofactors from the class of negative regulators[24]. This termination mechanism differs from the constitutive inhibition of the DEAD-box helicase eIF4AIII in the exon–junction complex, whereby MAGO, Y14 and MLN51 appear to enclose and stabilize the

RecA-like domains in the closed conformation[25]. Notably, CWC22 and NTC might contribute to the regulation of PRP2 by interacting with the RES complex and PPIL4, respectively. Finally, PRP2 dissociates during the spliceosome remodelling process driven by Aquarius.

Recombinant PRP2(137–1022) did not exhibit unwinding activity in vitro, either alone or in the presence of the cofactor GPKOW (Extended Data Fig. 7f–l and Supplementary Fig. 3). The same inactivity has been observed for the budding yeast counterpart Prp2p[26,27]. Whether this reflects a vital role of spliceosomes in activating translocation or an intrinsic inability of the helicase to use translocation for duplex separation remains unclear.

A question that is raised from these results is whether other splicing DEAH helicases also operate by translocation rather than winching[22,28,29]. PRP16 interacts with its targets in C complexes, the step I factors CCDC49 (Cwc25p in *S. cerevisiae*) and CCDC94 (Yju2 in *S. cerevisiae*), at the 3′ end of the branched intron before their release[16,30]. Reminiscent of the function of PRP2, PRP16 might translocate in a 3′-to-5′ direction to displace the intron-bound step I factors, thereby promoting the repositioning of the pre-mRNA substrate for exon ligation. Furthermore, although introns bound by PRP22 are not visible in cryo-EM maps, PRP22 slightly relocates between different C* complex variants[31], which suggests that translocation is possible.

Identifying PRP2^NTD as an accessory domain for multiple context-specific functions might provide suggestions on how NTDs function in other DEAH helicases. For instance, a fragment of PRP22^NTD (residues 393–427) inserts between the RT and linker domains of PRP8, which indicates that they have a role in anchoring the helicase downstream of the 3′ exon in the human C* and P spliceosomes[31–33]. Overall, the mechanism of action of PRP2 captured within B^AQR may provide a paradigm for other DEAH helicases and guide future investigations.

## Two helicases drive catalytic activation

An important implication of this work is that the B^act to B* transition, also known as catalytic activation, is driven by two helicases—PRP2 and Aquarius, acting on the branch site from downstream and upstream locations, respectively (Figs. 1b and 5b).

Arrested between the action of the two helicases, the B^AQR structure revealed how PRP2 operates at the B^act-to-B^AQR transition and implicates Aquarius as the helicase that promotes the conversion of B^AQR to B*. The requirement for two helicases is probably an adaptation for transferring the delicate branch duplex—which is often unstable owing to low complementarity[34,35]—to the catalytic centre while keeping it unaltered. Therefore, whereas PRP2 operates downstream of the branch site to unfasten the branch duplex from the constraints of the SF3B–RES complex, Aquarius acts upstream of the branch site to induce the removal of the remaining constraints and the relocation to the catalytic centre.

This two-step catalytic activation process could provide additional checkpoints to proofread and discard incorrectly selected introns, the length and sequence of which can vary substantially in metazoans[36,37]. This mechanism might also facilitate coordination between splicing and other intron-related events. Of relevance, Aquarius couples splicing to the biogenesis of intron-encoded box C/D small nucleolar RNP (snoRNP)[38], which suggests that it coordinates separation in time and space between snoRNP biogenesis and branch duplex transfer for splicing.

Aquarius binds the intron at position −40 to −33 upstream of BS-A in C complexes[38]. Despite the moderate resolution around Aquarius (around 6.2 Å), a comparison between B^AQR and C complexes showed a repositioning of the helicase, which suggests how it might operate (Extended Data Fig. 9). We have previously shown that Aquarius unwinds duplexes in vitro with apparent 3′-to-5′ polarity. As the mutant Aquarius(Y1196A) inactivates unwinding but not splicing, the relevance of this assay on splicing remains unclear[6]. However, putative translocation with 3′-to-5′ polarity might induce relaxation of the intron and perturb the contacts between the domains of Aquarius and adjacent subunits. This movement could propagate changes in interfaces between proteins interposed from Aquarius to the SF3B1–BS-A interface (primarily SF3A and SF3B subunits), which causes the release of BS-A and dissociation of the SF3A and SF3B complexes (Extended Data Fig. 9a–c).

Aquarius might also use 5′-to-3′ polarity, similar to the phylogenetically related SF1 helicase Upf1. By pulling the intron with this polarity, Aquarius could remotely eject BS-A from the primary pocket of SF3B1. Consequently, SF3B1 would switch from a loose to an open conformation, thereby restructuring HEAT repeats 16–20, which forms interfaces with PRP8, SF3B2 and SF3A2 (Extended Data Fig. 9d–g). Ultimately, the entire SF3A–SF3B complex would undock from the core components PRP8 and the snRNA U6, which accomplishes the conversion of B^AQR to B* and C complexes (Fig. 5b, Extended Data Fig. 8 and Supplementary Video 4).

The arrest of PRP2 in interactions with the molecular brake indicates that Aquarius starts acting after the complete translocation of PRP2. However, some chronological overlap of the actions of the two helicases cannot be excluded. For instance, Aquarius might change the tension in the intron to induce additional loosening of SF3B1 while PRP2 facilitates the complete liberation of the branch duplex.

Significantly, Aquarius and all molecular brake components—PPIL4, elements of PRP2^NTD and SKIP—are highly conserved among humans, plants and yeasts[39] (Supplementary Fig. 4). This conservation supports the view that Aquarius and the molecular brake have coevolved to complement PRP2 across all kingdoms of eukaryotes, which indicates that catalytic activation in two ATP-dependent steps might be a universal mechanism.

## Simplified catalytic activation in yeast

In contrast to fission yeast (*Schizosaccharomyces pombe*), budding yeast (*S. cerevisiae*) lacks Aquarius and the molecular brake, which raises the question of how catalytic activation can occur only by PRP2 (Prp2p in *S. cerevisiae*). Based on this work and a large body of published data about Prp2p, we propose a model for spliceosome remodelling in budding yeast (Fig. 5c).

Prp2p binds the 3′ tail of the intron, advancing in the 3′-to-5′ direction to disrupt interactions between SF3b1 (also known as Hsh155p) and the branch duplex while destabilizing SF3a, SF3b and RES complexes[19,23,40,41]. Consistent with the human model, this remodelling implies that Prp2p uses translocation rather than winching, as postulated in earlier models[19,40] (Fig. 5c). Spp2p, the G-patch orthologue of human GPKOW, assists these remodelling steps[1,5,27].

After destabilization of Res, we propose that Prp2p might transit into a B^AQR-like state, arriving at approximately 6 nucleotides downstream of the branch duplex. This scenario could explain the genetic interaction between the cold-sensitive allele *Prp2p*^Q548N (*PRP2*^Q721 in humans) and the SF3b1(D450G) or SF3b1(V502F) mutant[42,43]. The human counterparts SF3B1(D781) and SF3B1(L833) are in physical proximity to PRP2 only within B^AQR spliceosomes (Extended Data Fig. 4l).

Furthermore, the human B^AQR structure might explain the effect of Cwc22p(454–491) on the ability of Prp2p to remodel the B^act spliceosome[44]. The equivalent region of CWC22, the hook of PRP2^NTD (conserved in budding yeast) and the RES complex interact in B^AQR, which suggests that the role of Cwc22p is conserved in facilitating the function of the helicase[44] (Fig. 3b and Extended Data Fig. 4m–o). Notably, Prp2p^NTD is dispensable for viability[45], which suggests that neighbouring subunits might functionally compensate for its deletion.

In contrast to the human orthologue, Prp2p might advance beyond the B^AQR-equivalent time point, as there is no molecular brake to terminate translocation. The unrestricted Prp2p could extract BS-A from the primary pocket, which triggers the transition of SF3b1 from

the loose to the close conformation. In this way, Prp2p liberates the branch duplex and dissociates SF3a, SF3b and RES complexes, and the proteins Cwc24p and Cwc27p[19,23,41] (Fig. 5c). We suggest that this simplified mechanism is tolerated in budding yeast primarily because the high complementarity between the intron and U2 snRNA prevents its alteration by the unidirectional action of Prp2p.

## The complete cycle of SF3B1 transitions

Accumulating evidence outlines SF3B1 as a conformational switch adapted for the recognition and transfer of branch sites. This work provides new insight into the function and cyclic pathway of conformational transitions of SF3B1, mediated by RNA helicases (Extended Data Fig. 6d).

First, the ability of SF3B1 to switch conformations relies on two pockets—here referred to as primary and secondary pockets—that sequentially bind BS-A and PPT after the formation of the prespliceosome or release them in the reverse order during catalytic activation (Fig. 3d and Extended Data Fig. 6d). Both pockets are also conformational hinges, which thereby induce the progressive compaction or decompaction of the HEAT domain following the binding or release of cognate RNA elements. Consequently, SF3B1 transits stepwise to the open, half-closed and closed conformation after BS-A and PPT recognition during pre-A conversion to A complexes[46–48]. The open state facilitates the formation of the branch duplex by toehold-mediated strand invasion—a non-enzymatic mechanism captured in a human pre-A complex stalled using spliceostatin A[47].

The closed conformation persists until the B[act] complex, when PRP2 detaches the bound PPT from the secondary pocket, which then induces the loose conformation of SF3B1 within B[AQR]. The requirement of bound PPT for the closed conformation might explain the formation of B[act]-like spliceosomes on substrates bearing at least 16 nucleotides downstream of BS-A, whereas 6 nucleotides are insufficient[15]. Notably, the loose conformation differs from the half-closed one, which is probably due to the contacts between the PRP2[core] and HEAT repeats (410 Å$^2$ interfaces; Fig. 2b,c).

Next, Aquarius promotes the extraction of BS-A from the primary pocket and the release of SF3B, probably in the open conformation typical for the apo form[46]. The released SF3B might be recycled as a building block for the biogenesis of 17S U2 snRNPs assisted by the helicase SF3b125 (ref. 49), thus re-entering the splicing pathway. Overall, the structure of B[AQR] reveals the crucial importance of SF3B1 as a conformational switch in catalytic activation and how helicases can modulate the pathway of conformational transitions of SF3B1.

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

# Methods

## Cloning

Recombinant proteins were produced from codon-optimized synthetic genes (GeneArt, ThermoFisher Scientific). Full-length wild-type Aquarius and its K829A mutant were fused to a C-terminal 8×His-tag and cloned into the pFL vector backbone as previously described[6]. For co-expression of IBC subunits, the master, 4-protein construct ΔISCC, comprising SYF1 (also known as XAB2), ISY1, CCDC16 (also known as ZNF830) and PPIE (also known as CypE), was assembled by in vitro Cre−loxP recombination of the donor pSPL-CCDC16−PPIE and acceptor pFL-SYF1−ISY1 vectors[6]. The expression constructs were transformed into DH10MultiBacY *Escherichia coli* cells. The resulting bacmids were isolated using a High Pure Plasmid Isolation kit (Roche) and used for transfection into Sf9 (*Spodoptera frugiperda*) insect cells. Sf9 cells were transfected with the help of either X-tremeGENE HP (Roche) or FuGENE HD (Promega) reagents. Insect cell lines were not tested for mycoplasma contamination.

Human PRP2 (NCBI Reference Sequence identifier: NM_003587, transcript variant 1) and human PPIL4 (NCBI Sequence identifiers BC020986 and NM_139126.4) open reading frame clones were obtained from OriGene (RC202912) and Applied Biological Materials (373710120000), whereas human GPKOW and SKIP were codon-optimized for expression in insect cells and synthesized by GeneArt (ThermoFisher Scientific). All constructs were inserted by ligation-independent cloning into a modified pFastBac vector backbone in-frame with an N-terminal twin-StrepII affinity tag, which can be cleaved off with the HRV-3C protease. All recombinant constructs were verified by Sanger sequencing.

## Purification of recombinant proteins

Recombinant proteins were produced in insect cells using the Multi-Bac system[50]. The initial $V_0$ and $V_1$ baculovirus stocks were produced in Sf9 cells. Large-scale protein co-expression was conducted in High Five (Hi5) insect cells (*Trichoplusia ni*, BTI-TN5B1-4) through the combination of two $V_1$ stocks of baculovirus, coding for the C-terminal 8×His-tagged Aquarius (wild-type or mutant K829A) and the untagged ΔISCC complex, in 1:1.5 ratio ($V_1^{AQR}$:$V_1^{\Delta ISCC}$). The purification of the wild-type and the K829A dominant-negative IBCs was performed according to a previous protocol established in our laboratory, but with some modifications[6]. Insect cells expressing IBC were disrupted by sonication in lysis buffer (50 mM HEPES-NaOH, pH 7.5, 400 mM NaCl, 10% (v/v) glycerol, 5 mM 2-mercaptoethanol (2-ME), 30 mM imidazole and protease inhibitors (complete EDTA-free, Roche)) and the cell debris was pelleted by centrifugation at 18,000 r.p.m. at 4 °C for 1 h (A27-8×50 rotor; Thermo Scientific). The supernatant was loaded on a 5 ml HisTrap HP Ni(II)-chelating column (GE Healthcare/Cytiva) equilibrated with the lysis buffer. The Ni(II)-bound protein species were eluted using a linear 30−400 mM imidazole gradient in Ni(II) elution buffer (50 mM HEPES-NaOH, pH 7.5, 200 mM NaCl, 10% (v/v) glycerol and 5 mM 2-ME) and analysed by SDS−PAGE. The IBC-containing fractions were loaded on an anion-exchange Q Sepharose HP column (GE Healthcare/Cytiva) equilibrated in 20 mM HEPES-KOH, pH 7.5, 200 mM KCl, 10% (v/v) glycerol and 5 mM dithiothreitol (DTT). The bound IBC was eluted off the resin with a linear 0.2−1 M KCl gradient. The peak fractions, corresponding to the stoichiometric IBC, were pooled, concentrated to 10 mg ml⁻¹, aliquoted, flash-frozen in liquid nitrogen and stored at −80 °C.

Twin-StrepII-tagged human PRP2(137−1022), PPIL4 and GPKOW were expressed in Sf9 or Hi5 insect cells using recombinant baculoviruses, prepared as described above, in amounts sufficient to induce cell cycle arrest in 24 h. All recombinant proteins used in functional assays were purified at 4 °C using similar purification protocols. In brief, insect cells, expressing the target proteins, were briefly sonicated and lysed in lysis buffer (50 mM HEPES-KOH, pH 7.5, 150 mM KCl, 10% (v/v) glycerol, 0.2% (v/v) Triton X-100 RNase-free, 2 mM DTT and protease inhibitors (complete EDTA-free, Roche)), with the detergent added after sonication, and the cell debris was pelleted by centrifugation at 23,000 r.p.m. for 1 h in an A27-8×50 rotor (Thermo Scientific) or 40,000 r.p.m. (approximately 164,244$g$) in a type 70 Ti rotor (Beckman Coulter) for 45 min at 4 °C. The filtered supernatant was applied to a 5 ml Strep-Tactin XT 4Flow pre-packed column (IBA Lifesciences) or incubated in batch for 1 h with around 1 ml or 2.5 ml Strep-Tactin beads per 1 litre of Sf9 or Hi5 culture, respectively. The bound proteins were eluted by competition with 50 mM biotin in lysis buffer or with elution buffer containing 60 mM biotin (50 mM HEPES-KOH, pH 7.5, 100 mM KCl, 5% (v/v) glycerol, 1 mM EDTA, 2 mM DTT and 60 mM biotin). The Strep-Tactin eluates were concentrated and further purified by size-exclusion chromatography (SEC) on Superdex 200 HiLoad 16/600 200 pg (GE Healthcare/Cytiva) or Superdex 200 Increase 10/300 GL (GE Healthcare/Cytiva) equilibrated in 20 mM HEPES-KOH, pH 7.5, 150 mM KCl, 10% (v/v) glycerol and 2 mM DTT. Alternatively, the PRP2(137−1022) Strep-Tactin eluates were purified on Q Sepharose HP (GE Healthcare/Cytiva) and eluted from the 5 ml anion-exchange column using a linear 0−30% gradient formed over 40 ml between buffer A (20 mM HEPES-KOH, pH 7.5, 100 mM KCl, 5% (v/v) glycerol and 0.5 mM DTT) and buffer B (20 mM HEPES-KOH, pH 7.5, 1 M KCl, 5% (v/v) glycerol and 0.5 mM DTT). With the exception of PPIL4, all proteins used in functional assays comprised the N-terminal Twin-StrepII tag. The Twin-StrepII tag of PPIL4 was removed by on-bead cleavage with the HRV-3C protease, and the untagged protein was purified by SEC on Superdex 200 Increase 10/300 GL (GE Healthcare/Cytiva). Purified proteins were concentrated by ultrafiltration to their stock concentration (PRP2(137−1022), around 19−22 μM; GPKOW, around 165 μM; untagged PPIL4, around 47 μM; tagged PPIL4, about 67 μM), frozen in liquid nitrogen and stored at −80 °C or used directly in assays.

## Reconstitution of PRP2−GPKOW and PRP2−PPIL4 complexes

Using a minimal in vitro system, we first tested direct interactions and stable formation of complexes between PRP2(137−1022) and its cofactors GPKOW and PPIL4 as subjected to SEC. All SEC analyses were carried out in SEC buffer (20 mM HEPES-KOH, pH 7.5, 100 mM KCl, 1.5 mM MgCl₂ and 5% (v/v) glycerol). The PRP2 complexes were reconstituted in vitro by mixing around 30 μg recombinant helicase with a 5-fold excess of cofactor (PPIL4 or GPKOW) in a 100 μl reaction volume. The samples were then incubated for 1 h on ice and applied to a Superdex 200 Increase 10/300 GL column (GE Healthcare/Cytiva) run at 0.5 ml min⁻¹ using an Äkta Go system (Cytiva). SDS−PAGE analyses of the SEC fractions showed that in both cases, the peak profile of PRP2(137−1022) shifted to early fractions in the presence of a cofactor (fraction 9 in the presence of PPIL4 and fraction 7 in the presence of GPKOW) compared with the protein alone (fraction 11), which indicated the formation of a helicase−cofactor complex. As independent means of probing the direct interaction of PRP2(137−1022) with PPIL4, we co-expressed the two splicing factors in Sf9 insect cells and affinity purified their complex from baculovirus-infected cultures. The cultured cells were lysed in lysis buffer (50 mM HEPES-KOH, pH 7.5, 150 mM KCl, 5% (v/v) glycerol, 2 mM DTT, 2 mM MgCl₂, 0.1% (v/v) Triton X-100 and protease inhibitors (complete EDTA-free, Roche)) and the expressed factors were captured on Strep-Tactin XT 4FLOW affinity beads (IBA Lifesciences). The protein samples were eluted by competition with biotin in elution buffer (50 mM HEPES-KOH, pH 7.5, 150 mM KCl, 5% (v/v) glycerol, 2 mM DTT, 2 mM MgCl₂ and 60 mM biotin), concentrated by ultrafiltration to about 500 μl and subjected to SEC in SEC buffer. As for the in vitro assembled PRP2(137−1022)−PPIL4 complex, the complex prepared from insect cells peaked in fraction 9 when eluting from the Superdex 200 column. Overall, this result shows that the limited interaction interface between the NTD of PRP2 and PPIL4, observed in the B^AQR cryo-EM structure, promotes the stable recruitment of the helicase cofactor in a RNA-independent manner. The PRP2(137−1022)−PPIL4 complex used in helicase activity assays was prepared by insect cell co-expression,

purified in two steps, as described above, and concentrated to about 14.5 µM. The PRP2(137–1022)–GPKOW complex used in biochemical assays was assembled in vitro, purified by SEC from GPKOW excess and concentrated to about 40.3 µM. In vitro reconstitution of the complex between PRP2(137–1022) and PPIL4 was performed two times using two independent protein preparations. The PRP2(137–1022)–GPKOW complex was assembled in vitro at least three times.

## Preparation of the nuclear extracts active in splicing

HeLa S3 cells, tested for mycoplasma, were obtained from GBF (Helmholtz Centre for Infection Research). The nuclear extract active in splicing was prepared according to the standard protocol from Dignam and used as described[13]. Cells were grown in a 30 litre fermenter (Applikon Biotek) to a density of $6.5 \times 10^6$ cells per ml in DMEM/F12 (1:1) medium supplemented with 5% (v/v) newborn calf serum. After collection by centrifugation for 10 min at 2,000 r.p.m. in an $8 \times 2,000$ ml BIOS rotor (Thermo Scientific), the cells were washed twice with cold 1× PBS buffer. The cell pellet was re-suspended in MC buffer (10 mM HEPES-KOH, pH 7.6, 10 mM potassium acetate, 0.5 mM magnesium acetate, 0.5 mM DTT and 2 tablets of protease inhibitors (complete EDTA-free, Roche) per 50 ml of the buffer). After 5 min of incubation on ice, the cells were lysed with 18 strokes of a Dounce homogenizer at 4 °C. The nuclei were pelleted for 5 min at 10,000 r.p.m. in a F14-14×50cy rotor (Thermo Scientific) and were further lysed in Roeder C buffer (20 mM HEPES-KOH, pH 7.9, 0.2 mM EDTA, pH 8.0, 25% (v/v) glycerol, 420 mM NaCl, 1 mM $MgCl_2$, 0.5 mM DTT and 0.5 mM PMSF) by 20 strokes of a Dounce homogenizer at 4 °C. The mixture was stirred slowly for 40 min at 4 °C and centrifuged at 12,300 r.p.m. in a F14-14×50cy rotor. The supernatant corresponding to the active nuclear extract was aliquoted and flash-frozen in liquid nitrogen and stored at −80 °C.

## In vitro pre-mRNA splicing

A typical splicing reaction contained the following components: 20% (v/v) non-dialysed HeLa nuclear extract, 3 mM $MgCl_2$, 2 mM ATP, 20 mM creatine phosphate and 10 nM $m^7G$-capped MINX-3×MS2 pre-mRNA, fluorescently body-labelled with cyanine 5-uridine-5′-triphosphate (Cy5-UTP, Enzo). To monitor splicing of the Cy5-labelled pre-mRNA substrate, the splicing reactions were incubated for 15, 30, 60, 90 and 120 min at 30 °C (Extended Data Fig. 1c). To assess the effect of recombinant IBC(K829A) on the assembly of the spliceosome, the splicing reaction was supplemented with IBC(K829A) to a final concentration of 0.45 µM and pre-incubated for 20 min at 30 °C before the addition of the pre-mRNA substrate and ATP (Extended Data Fig. 1c). In all cases, RNA was recovered by phenol–chloroform–isoamyl alcohol extraction followed by ethanol precipitation. The recovered RNA samples were analysed on denaturing urea polyacrylamide gels (14%, 0.5× TBE). The fluorescently labelled pre-mRNA substrate, splicing intermediates and the final products were detected by in-gel fluorescence using a Typhoon FLA 9500 imaging system (GE Healthcare).

## Affinity purification of the human $B^{AQR}$ spliceosome

Spliceosomal complexes stalled with recombinant IBC(K829A) were assembled in vitro on non-labelled $m^7G$-capped MINX-3×MS2 pre-mRNA, synthesized by T7 RNA polymerase run-off transcription. Before initiating the reaction, pre-mRNA substrate (used at 10 nM, final concentration) was incubated with a 20-fold molar access of MBP-MS2 fusion protein for 30 min at 4 °C. Before the addition of ATP and pre-mRNA-MBP-MS2, the large-scale splicing reactions were supplemented with 0.45 µM recombinant IBC(K829A) complex and incubated for 20 min at 30 °C. Therefore, a typical preparative splicing reaction, used from cryo-EM sample preparation, comprised (final concentrations) the following: 20 mM HEPES-KOH, pH 7.9, 2 mM ATP, 20 mM creatine phosphate, 3.2 mM $MgCl_2$, 84 mM NaCl, 5% (v/v) glycerol, 20% (v/v) non-dialysed HeLa nuclear extract, 10 nM MINX-3×MS2, 200 nM MBP-MS2 and 40 µM IBC(K829A).

To enable the spliceosome assembly, the splicing mixture was slowly stirred for 120 min at 30 °C in a water bath. The non-incorporated pre-mRNA was cleaved by DNA-directed endogenous RNase H digestion for 20 min at 30 °C using a 30-fold molar excess of the cmd42 (5′-TCTTACCGTTCG-3′) and cmd43 (5′-CGGGTTTCCGAT-3′) antisense DNA oligonucleotides. To prevent precipitation of assembled spliceosomes, NaCl was slowly added to the splicing reaction to a final concentration of 120 mM. The aggregates were removed by centrifugation for 10 min at 12,300 r.p.m. in a F14-14×50 rotor. The supernatant was then applied onto a pre-packed 5 ml MBPTrap HP column (GE Healthcare/Cytiva) at 1 ml min$^{-1}$, washed with 20× column volumes of G120 buffer (20 mM HEPES-KOH, pH 7.9, 1.5 mM $MgCl_2$ and 120 mM KCl) and the spliceosomal complexes were eluted with G120 buffer supplemented with 3 mM maltose. The peak fractions were analysed by denaturing 4–12% NuPAGE gels (Life Technologies). The gels were stained with SYBR Gold (Invitrogen) and Coomassie to detect RNA and protein species, respectively. The elution fractions containing IBC(K829A) spliceosomes ($B^{AQR}$) were loaded onto a linear 10–30% (w/v) glycerol gradient prepared in G120 buffer and centrifuged at 20,500 r.p.m. for 16 h at 4 °C in a TST 41.14 rotor (Kontron). The gradients were divided into 23 fractions (500 µl each) and manually collected from the top. Peak fractions (14, 15 and 16) from 3 different gradients were pooled and crosslinked with 0.1% (v/v) glutaraldehyde (Electron Microscopy Sciences) for 1 h on ice. The unreacted glutaraldehyde was quenched with 100 mM aspartate (final concentration). The crosslinked $B^{AQR}$ was then incubated for 2 h on ice and concentrated to the sample absorbance at 280 nm of 0.6. The buffer was exchanged by ultrafiltration to the sample buffer (20 mM HEPES-KOH, pH 7.9, 120 mM NaCl, 1.5 mM magnesium acetate and 2% (w/v) glycerol) using an Amicon 50 kDa MWCO (Millipore) spin concentrator and used directly for cryo-EM grid preparation.

## Preparation of cryo-EM grids

Volumes of 4 µl of concentrated sample were applied to one side of glow-discharged UltrAuFoil 200 R2/2 grids (Quantifoil) in a Vitrobot Mark IV (FEI) operated at 4 °C and 100% humidity. The grids were blotted for 2 s with a blotting force of 5 and immediately frozen by plunging into liquid ethane cooled by liquid nitrogen.

## Cryo-EM data acquisition and image analysis

The electron micrographs for all datasets were acquired on an FEI Titan Krios G2 transmission electron microscope operated at 300 keV in EFTEM mode, equipped with a Quantum LS 967 energy filter (Gatan), zero loss mode, 30-eV slit width and a K2 Summit direct electron detector (Gatan) in counting mode. Automated data acquisition for dataset 1 (untilted) and dataset 2 (tilted, 25°) was performed using EPU software (Thermo Fisher) at a nominal magnification of ×130,000 (1.05 Å per pixel (Å px$^{-1}$)). Micrographs for these two datasets were collected as 40-frame movies at a dose rate of around 5 e$^-$ Å$^{-2}$ s$^{-1}$ over 9 s, which resulted in a total dose of about 45 e$^-$ Å$^{-2}$.

The two cryo-EM datasets of $B^{AQR}$, collected at 0° and 25° tilt angles, were separately preprocessed on the fly with Warp (motion correction, CTF estimation and dose weighting), and particles were picked using a retrained convolutional neural network[51]. Each set of Warp-extracted particles was then subjected to three parallel 2D classification runs in cryoSPARC (v.0.65) using 50 classes and applying a class uncertainty factor of 1.5. The good classes were selected from the three runs, and all particles in these good classes were merged (while removing duplicates).

Initial attempts at processing the cryo-EM images in cryoSPARC revealed that this spliceosome complex exhibited a more structurally rigid core, which continued with more dynamic and poorly resolved peripheral regions (Extended Data Fig. 2). Hence, we systematically tested different EM data processing routines with the aim of improving the more peripheral densities of the spliceosome and to computationally resolve its structural heterogeneity. In brief, the Warp-picked,

combined $B^{AQR}$ particle images (734,691 particles) were re-extracted in Relion 3.1 using a box size of 640/640 px (672 Å/672 Å) and then 2× binned before being subjected to 2D classification with the 'Ignore CTFs until the first peak' option switched on (Extended Data Fig. 2). The resulting 'good' particles subset (160,650 particles) was then refined in 3D using a 60 Å low-passed human $B^{act}$ map as a reference volume (Electron Microscopy Databank identifier EMD-4236)[13] and a spherical mask. The obtained consensus map of the complex (map M1) was then used as a reference in a global 3D classification round with 10 classes starting from the initial pool of $B^{AQR}$ particles (Extended Data Fig. 2). The cleaned subset of $B^{AQR}$ particles was then subjected to another round of global 3D classification with 8 classes and a 20 Å resolution limit. This second-round of 3D classification enabled us to resolve the two major compositional states of the complex (Extended Data Fig. 2), termed state A (29.2% particles) and state B (3.4% particles). In the state B complex, stronger density was observed for the PRP19 helical bundle and the step two splicing factor PRP17. Conversely, these map regions appeared to be poorly resolved in the state A complex, which is probably due to their flexibility and low occupancy, as observed for a previously described $B^{act}$ complex[13]. In both cases, however, the peripheral building blocks of the complex (the IBC, U2 3′ core and the BRR2 helicase) were less resolved than the central part of the complex. To further improve the density of the state A map, we performed an additional global 3D classification without image alignment with 8 classes and a 20 Å resolution limit and then re-extracted and re-centred the $B^{AQR}$ particle images at their original sampling rate (1.05 Å px$^{-1}$) in a 520 px box. The 3D refinement of this subset of particles (179,552 particles) with a soft mask was then followed by CTF refinement (per particle defocus, per micrograph astigmatism) and an additional round of classification in 2D. The subsequent refinement in 3D of this final subset of particles (146,157 particles) with loose or tight soft masks around $B^{AQR}$ resulted in map 2 and map 3 of the complex that reached global, gold-standard Fourier shell correlation resolutions of about 3.1 Å and 2.9 Å, respectively (Extended Data Figs. 2 and 3). Using these well-resolved consensus maps, we modelled the translocating PRP2 helicase, including its extended NTD domain, as well as the remodelled SF3B complex, and two out of the four $B^{AQR}$ PPIases (that is, PPIL2 and PPIL4).

Starting from the particle sets, which led to the higher resolution core maps, we also carried out local 3D classifications (Extended Data Fig. 4). Thus, by applying a mask covering the overall $B^{AQR}$ volume, we subjected these particles to 3D classification without image alignment with 6 classes. Particles exhibiting more pronounced densities for the U2 3′ core module and the IBC module (27,719 particles) were subsequently re-extracted and refined in 3D with soft and spherical masks. These alternative $B^{AQR}$ maps enabled the rigid body placement of the U2 3′ core and IBC modules at the periphery of $B^{AQR}$. Local classifications were also performed on the particle set assigned to state B of the complex. In this case, a soft mask was applied to the region of the spliceosome where the BRR2 helicase resides, which was poorly resolved in the consensus map. The state B particle images were then subjected to 3D classification without image alignment and 4 classes. A minority subset of the overall particle set (12,395 particles) appeared to contain a more ordered BRR2, with the helicase being generally destabilized in $B^{AQR}$ cryo-EM images, which is probably because of the substantial change to the conformation of SF3B1 induced by the translocation of PRP2. These particles were then re-extracted and refined in 3D to obtain the complete map of the complex (map M4), which now enclosed the U2 3′ core, the IBC module, the PRP19 helical bundle, the BRR2 helicase and the splicing factor PRP17.

### Density assignment, model building and refinement

To enable model building, the $B^{AQR}$ core maps (Supplementary Fig. 2) were sharpened using DeepEMhancer[52] or locally scaled with LocScale[53]. The quality of the cryo-EM density map at the core of $B^{AQR}$ enabled

careful model building and side-chain assignment, whereas modelling of the more solvent-exposed map regions was restricted to backbone tracing and rigid-body docking of known structures and computational models. Initial interpretation of the $B^{AQR}$ maps was facilitated by the available models of human $B^{act}$ complexes obtained in several states (PDB identifiers 6FF4, 6FF7 and 5Z57). Thus, the model building of the $B^{AQR}$ complex was initiated by the docking of the human $B^{act}$ models into the consensus maps (maps 2, 3 and 4) of the complex, followed by manual, residue-by-residue model adjustment in Coot[54] and refinement with phenix.real_space_refine[55]. Although filtered, locally scaled maps were used for model interpretation, the real space refinement of the $B^{AQR}$ model was carried out exclusively against the original, unsharpened maps. The higher local resolution of our maps enabled us to improve and correct some of the available models for the $B^{act}$ core region. After initial placement and refinement, several new (or reconfigured) $B^{AQR}$ density regions were observed compared with previous $B^{act}$ complexes. The globular density element identified in the U2 snRNP region of the complex and surrounded by the HEAT domain of SF3B1 was assigned to the helicase domain of PRP2 (residues 388–1017). We modelled the helicase in an open state, consistent with it being trapped in a post-translocation state, and identified seven intron nucleotides accommodated by its RNA-binding tunnel. Compared with a previously published yeast $B^{act}$ structure (PDB identifier 7DCO), the helicase domain of PRP2 (PRP2$^{core}$) was no longer positioned on the convex side of SF3B1$^{HEAT}$, but translocated along the intron towards the branch helix. The PRP2-bound RNA formed a continuous density stretch with the intron strand of the U2–BS duplex, showing that the helicase had translocated in a 3′-to-5′ direction from its position in the $B^{act}$ complex. Supporting this model, density for the RES complex subunits RBMX2 and SNIP1, which bind intron regions (or are located close) downstream of the branch helix, could not be observed in $B^{AQR}$, whereas only a short helical region belonging to the BUD13 subunit (residues 530–557) was identified in the proximity of the MA3 domain of CWC22. Besides modelling the helicase domain of PRP2, we also built de novo a large part of the PRP2$^{NTD}$ that was missing in previous $B^{act}$ structures. The N-terminal most α-helical region (the pin, residues 161–193) was positioned at the periphery of the helicase domain, where it established interfaces with the RecA1, WH and HB domains of the helicase; two other long helices (the clip, residues 223–256 and residues 264–296) were engaged in tight interactions with the PPIL4 linker region and the long helix of SKIP (residues 286–340), respectively. The latter PRP2$^{NTD}$ helix extended further towards PRP8 and finally reached CWC22.

The new density of PRP2 in $B^{AQR}$ coincides with a significant change to the curvature of SF3B1$^{HEAT}$, with its N-terminal HEAT repeats no longer in contact with the BS–U2 helix. However, the density of BS-A was still observed in the SF3B1–PHF5A binding pocket, with the hinge region of SF3B organized as in the closed state. Consistently, the SF3A2 and SF3A3 matrin-type zinc-finger domains were organized and positioned as in the $B^{act}$ complex and still engaged the 5′-end of the U2 snRNA. We modelled SF3B1 by individually docking its consecutive HEAT repeats and manually adjusting their fit to the map. The resulting structure of the reconfigured SF3B1 differed from all other known conformations of SF3B1$^{HEAT}$. Because of the PRP2-induced reconfiguration of SF3B1$^{HEAT}$, its SF3B6 and SRRM1 binding partners were destabilized from their earlier locations on the N-terminal side of the HEAT superhelix and, compared with $B^{act}$, their densities were no longer observed in $B^{AQR}$.

Facing the PRP2 density element and following the RNA density exiting the helicase cassette, a new small globular domain appeared to be recruited to $B^{AQR}$. We assigned it to the C-terminal RRM region of PPIL4 on the basis of its continuity, through an α-helical density, to the predicted cyclophilin-type PPIase domain that interacts with the long helix of SKIP(286–340) and proteomics analysis. A similar SKIP-bound PPIase density has previously been observed in the later C complex (PDB identifier 5yzg), at an almost identical position; in the published C complex model (PDB 5YZG), the authors assigned it to the

PPIase domain of the PPIase PPIG[30]. The latter lacks a RRM domain and is more abundant in C complexes, but only in trace amounts in B[AQR]. The other PPIase identified in the B[AQR] proteome, which comprises a PPIase and an RRM domain, is PPIE, an IBC module component located at the periphery of the spliceosome.

In addition to PPIL4 and PPIE, we modelled two other PPIases: PPIL1, interacting with and, probably, stabilizing the PRP19 helical bundle onto the B[AQR] core and PPIL2. The latter PPIase adopted an extended conformation with its N-terminal tandem U-box motifs, interacting with SNU114 and separated from its C-terminal PPIase domain by about 90 Å. An ordered linker region (residues 234–266) connected the two PPIL2 moieties, as previously observed in a B[act] cryo-EM structure[5]. We did not observe density for the PPIase domain of CWC27, which is consistent with it being destabilized by the propagated changes induced by the remodelling of SF3B1. However, its interacting partner CWC24 was still present in B[AQR], with densities observed for both its zinc finger motif (residues 190–238), sequestering the 5′SS and its C-terminal moiety (residues 262–309) bound to the BPB domain of SF3B3. The final model of the complex (Supplementary Data 2), including its more dynamic peripheral modules, consisted of four RNA molecules (the U2, U5 and U6 snRNA, and the MINX pre-mRNA substrate) and 45 individual polypeptide chains, among which 3 are splicing helicases (PRP2, BRR2 and Aquarius) and 4 are PPIases (PPIL1, PPIL2, PPIL4 and PPIE).

## Morphing and generation of movies

The PRP2 helicase translocation trajectory was generated using ChimeraX (v.1.3), with the morph functionality starting from the B[act] state of the SF3B module (that is, SF3B1 in a closed conformation with PRP2 positioned on the convex side of SF3B1[HEAT]). The NTD domains of SF3B1 (SF3B1[NTD]) and PRP2 (PRP2[NTD]) were not considered in the morphing analysis, and the U2 sequence was limited to the U2 stem-loop IIa structure and the BS-interaction region. The initial (B[act]) and final (B[AQR]) trajectory snapshots were aligned using the PHF5A subunit of SF3B as a reference before morphing. The RNA-bound state of PRP2 in B[act] was modelled based on the yeast B[act] structure[5]. Supplementary Videos 1–4 were generated using ChimeraX (v.1.3), and Supplementary Video 5 was generated using PyMol.

## Helicase assays

The helicase assays were performed as previously described for other splicing helicases[6,56–59]. To assess the ability of human PRP2 to unwind the U2–BS helix in the presence of its different cofactors, we performed in vitro helicase assays on fluorescently labelled model substrates comprising a perfect double-stranded RNA duplex followed by a single-stranded 3′ overhang. The unwinding activity of PRP2 activity was monitored using either a gel-based readout[6], in which the single-stranded Cy5-labelled product is separated from the helicase substrate on a native polyacrylamide gel, or by recording the time course of decrease in the fluorescence of the substrate[58] as the duplex is unwound. In this case, the dual-labelled RNA strand, displaced because of the helicase activity, forms an intramolecular hairpin that brings in proximity the terminal Cy5 probe and its spectrally overlapping dark quencher (BHQ-2), thereby leading to fluorophore emission quenching and fluorescence decay.

The helicase substrate for the gel-based assay was prepared by mixing 30 µM Cy5-labelled strand (5′-CACCAGCUCCGUAGGCGC-Cy5-3′) with 45 µM unlabelled RNA oligonucleotide (5′-**GCGCCUACGGAGCU GGUG**GCGUAGGCGCAAAAAAAAAAAAAAAAAAAAAAA-3′, the complementary region is shown in bold) in 20 mM HEPES-KOH, pH 7.5. The RNA substrate used in the fluorescent-based helicase assay[58] was prepared by mixing in a similar molar ratio the dual-labelled RNA oligonucleotide (5′-Cy5-GCGCCUACGCCACCAGCUCCGUAGGCGC-BHQ-2-3′) with the unlabelled strand (5′-**GCGCCUACGGAGCUGGUGGCGUA GGCGC**AAAAAAAAAAAAAAAAAAAAAAA-3′, the complementary region is shown in bold) in RNA annealing buffer (6 mM HEPES-KOH, pH 7.5,

50 mM KCl and 0.2 mM MgCl$_2$). In both cases, the single-stranded RNA oligonucleotides were annealed by sequential incubation at 95 °C for 2 min, then at 80 °C for 10 min and then slowly cooled down to room temperature and stored on ice. The RNA and DNA oligonucleotides used in the helicase assays were obtained from IDT (Integrated DNA Technologies) or Microsynth.

In a typical gel-based helicase assay, 50–100 nM fluorescently labelled RNA substrate was mixed in 20 µl on ice with increasing concentrations of PRP2 constructs or PRP2 complexes (1–10 µM) in helicase assay buffer (20 mM HEPES-KOH, pH 7.5, 50 mM KCl, 2 mM MgCl$_2$, 0.1 mg ml$^{-1}$ BSA (NEB), 0.08 U µl$^{-1}$ RNasin (Promega), 5% (v/v) glycerol and 0.5 µM competitor DNA (5′-GCGCCTACGGAGCTGGTG-3′), final concentrations). The GPKOW cofactor was added in a 5-fold molar excess over PRP2 (137–1022) or PRP2(137–1022)–PPIL4. After pre-incubation for 10 min at 37 °C (or 20 min at 30 °C), the unwinding reaction was initiated by the addition of 2 mM ATP, and the samples were incubated for 1 h at 37 °C (or 1 h at 30 °C). The reactions were stopped by the addition of 5 µl quenching buffer (0.83 mM Tris-HCl, pH 7.6, 5 mM EDTA, 5% (v/v) glycerol, 0.0025% bromophenol blue, 0.0025% xylene cyanol FF and 0.04 U µl$^{-1}$ proteinase K (NEB), final concentrations) and incubated for 20 min at 37 °C. RNA duplex unwinding was assessed on a 14% polyacrylamide native gel prepared in 1× TBE or 1× Tris-glycine buffer and was run in 0.5× TBE (or 1× Tris-glycine) at 100 V for approximately 90 min at room temperature. The yeast Prp22p helicase, used as a positive control, was prepared as previously described[27]. The RNA gels were scanned at the Cy5 excitation peak using an iBright 1500 imaging system (Invitrogen). The gel-based helicase assays were repeated at least two times.

The molecular beacon helicase assays were performed using a Fluorolog spectrofluorometer (Horiba). In brief, 200 nM dual-labelled substrate was mixed with 1–5 µM purified helicase in reaction buffer (20 mM HEPES-KOH, pH 7.5, 50–150 mM KCl and 2 mM MgCl$_2$) in the absence of ATP and preincubated for at least 10 min at room temperature. As for the gel-based unwinding assays, GPKOW was added in a 5-fold molar excess over PRP2(137–1022), whereas PPIL4 was added in a 3-fold molar excess. The reactions were subsequently transferred to a 1.5 mm ultra-microcuvette (105.252-QS, Hellma) and equilibrated at the measurement temperature (25 °C or 30 °C) in the spectrometer. The unwinding reaction was initiated by the addition of 2 mM ATP (final concentration), and the decay in Cy5 fluorescence was immediately recorded at an interval time of 1 s and an integration time of 0.1 s. The Cy5 fluorescent probe was excited at 645 nm (3 nm slit width) and its emission was measured at 667 nm (3 nm slit width) and corrected for detector noise. Data were analysed and plotted using OriginPro 2020 (9.7.0.188). The fluorescence-based helicase assay was repeated at least three times. Three independent preparations of PRP2(137–1022) were tested in these assays.

## Electrophoretic mobility shift assay

To investigate the RNA binding activity of the different PRP2 complexes tested in the unwinding assays, 50 nM fluorescent substrate was mixed with increasing concentrations of helicase or helicase cofactor samples (0.1–8 µM). The 20 µl binding reactions were prepared in RNA binding buffer (20 mM HEPES-KOH, pH 7.5, 50 mM KCl, 2 mM MgCl$_2$, 0.1 mg ml$^{-1}$ BSA (NEB) and 5% (v/v) glycerol, final concentrations) and incubated for 30 min at room temperature before being loaded on a 5% polyacrylamide native gel prepared in 1× Tris-glycine buffer. The native RNA gel was pre-run at 40 V for 30 min at room temperature and run then at 60 V in 1× Tris-glycine. The gels were scanned at the Cy5 excitation peak. The RNA binding assays were repeated at least two times, and similar results were obtained.

## Reporting summary

Further information on research design is available in the Nature Portfolio Reporting Summary linked to this article.

## Data availability

The coordinate files have been deposited into the PDB with the identifiers 7QTT (high-resolution core) and 8CH6 (overall composite model). The cryo-EM maps have been deposited into the Electron Microscopy Data Bank with the identifiers EMD-14146 (high-resolution core) and EMD-16658 (overall reconstruction). The mass spectrometry protein composition and model building of the B$^{AQR}$ spliceosomes are available at FigShare (https://doi.org/10.6084/m9.figshare.22047275).

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

**Acknowledgements** We thank C. Richardson (The Institute of Cancer Research, London) for technical support in computation; M. Raabe, S. König and A. Chernev (Max-Planck Institute, Göttingen) for mass spectrometry analysis; S. Bessonov (Max-Planck Institute, Göttingen) for technical advice in splicing biochemistry; and A. Hoskins (University of Wisconsin-Madison, USA) for critical reading of the manuscript. This work was supported by the German Research Foundation (grant DFG PE 2079/2-2 and DFG PE 2079/4-1), the Wellcome Trust (220300Z/20/Z) and the Institute of Cancer Research.

**Author contributions** J.S. established purification of the spliceosomes, prepared grids and collected cryo-EM data, processed cryo-EM data, reconstructed initial maps and built initial models by docking. C.C. performed advanced processing of cryo-EM data, which resulted in the final deposited maps. C.C. built and refined the structural models. J.S. and C.C. purified recombinant proteins. C.C. cloned the expression constructs, performed the interaction assays and helicase assays with recombinant proteins. C.D. assisted with cryo-EM data collection and early data processing. H.U. performed mass spectrometry analyses. C.C. and V.P. wrote the paper, with input from J.S. and C.D. V.P. supervised the research.

**Competing interests** The authors declare no competing interests.

**Additional information**
**Correspondence and requests for materials** should be addressed to Vladimir Pena.

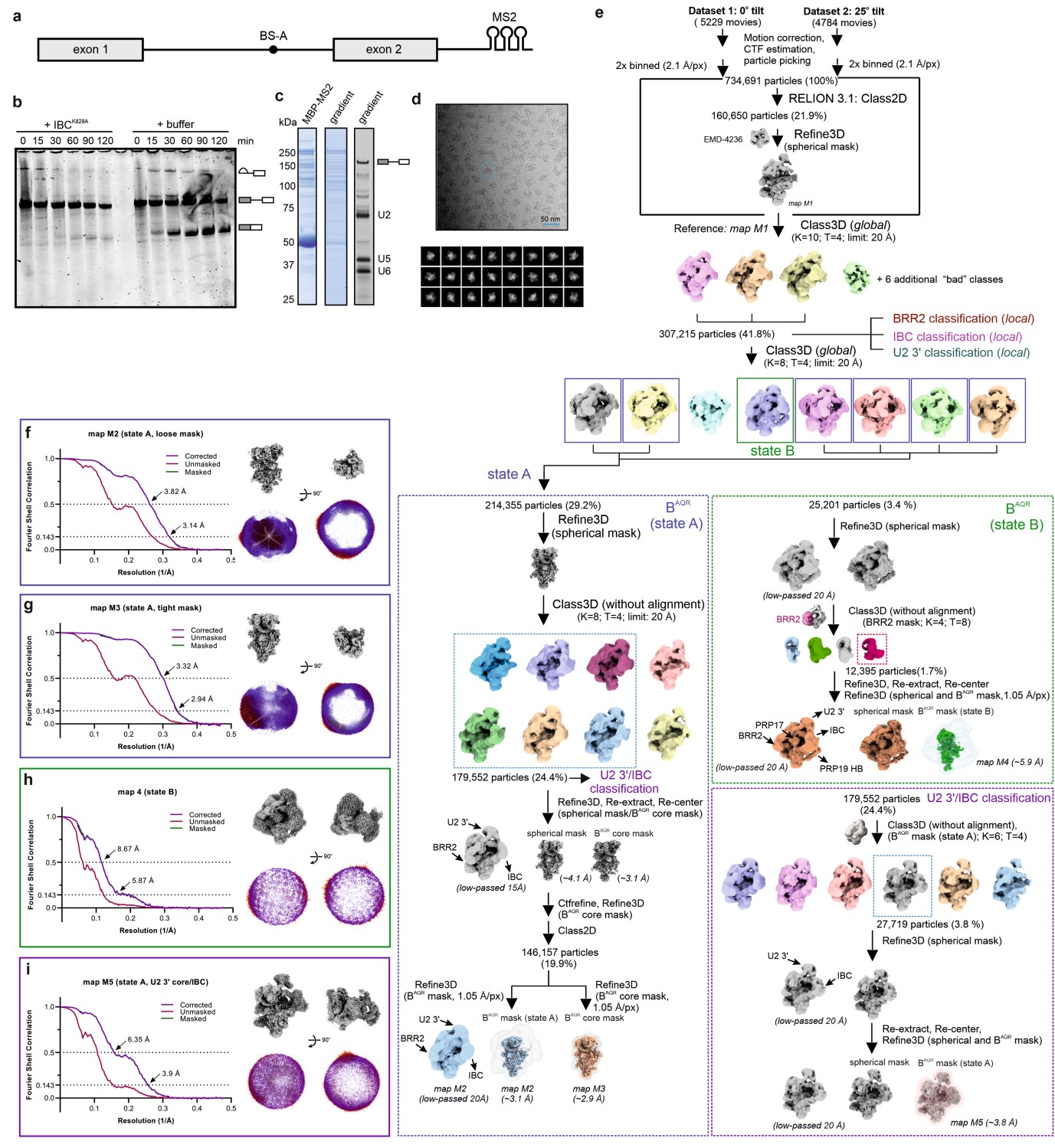

**Extended Data Fig. 1 | Purification, biochemical analysis and cryo-EM image-processing workflow of B^AQR. a**, Schematic depicting the MINX pre-mRNA substrate, fused to three downstream MS2 aptamers, which was used to assemble the B^AQR spliceosome in the HeLa nuclear extract. **b**, Pre-mRNA splicing is arrested by incorporating the recombinant dominant-negative IBC/Aquarius^K829A in the human spliceosomes. Typical *in vitro* splicing reactions of the model MINX pre-mRNA substrate were set up in the presence (+IBC^K829A) or absence (+buffer) of the recombinant complex added *in trans*, in excess over the endogenous IBC. The reactions were stopped after 0–120 min and the Cy5-labeled RNA substrate was extracted, analyzed on a denaturing polyacrylamide gel, and visualized by in-gel fluorescence. The experiment was repeated at least three times, with sample resulting from distinct preparations, with similar results. **c**, Protein and RNA composition of the purified B^AQR complexes. The SDS-PAGE gels were stained with Coomassie (left) and SYBR Gold (right). The purification was repeated at least 3 times, with similar results. For gel source data, see Supplementary Fig. 1. **d**, Representative cryo-EM micrograph of the stalled B^AQR spliceosome and typical reference-free 2D class averages of B^AQR particle images. We collected two independent datasets from two different sample preparations **e**, Cryo-EM data processing schematic showing the most significant steps of the computational analysis. **f–i**, Global resolution of key B^AQR maps (see also Extended Data Fig. 2). The FSC (Fourier Shell Correlation) analysis was carried out in RELION 3.1 (left). The angular distribution (right) of the B^AQR particles contributing to the map shown on top is depicted in two different orientations, with the red color indicating a higher number of particles at a given projection angle.

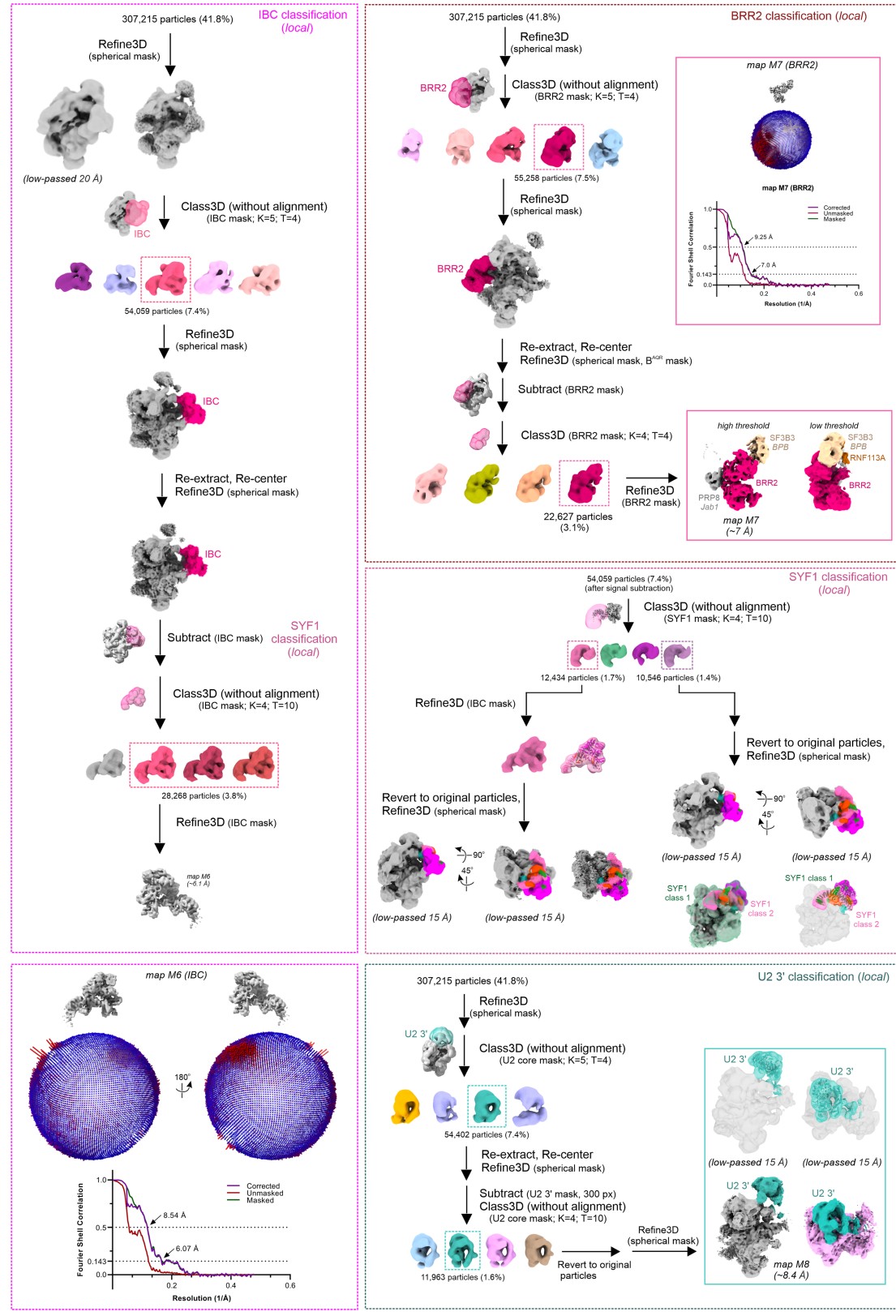

**Extended Data Fig. 2 | Cryo-EM image-processing by local classification of the peripheral modules of the B^AQR complex.** Cryo-EM data processing schematic showing the most significant steps of the computational analysis, the FSC (Fourier Shell Correlation) analysis of the obtained focused maps, and the angular distribution of the particles contributing to the final volumes.

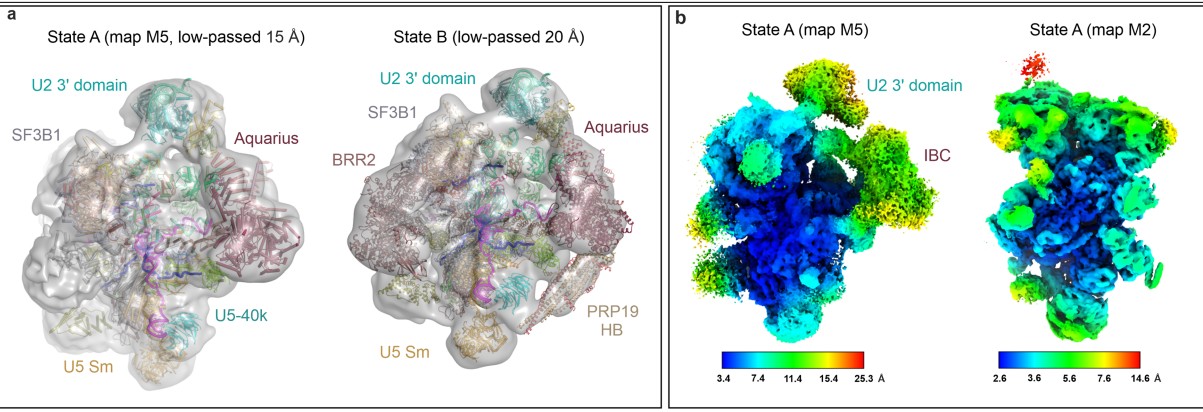

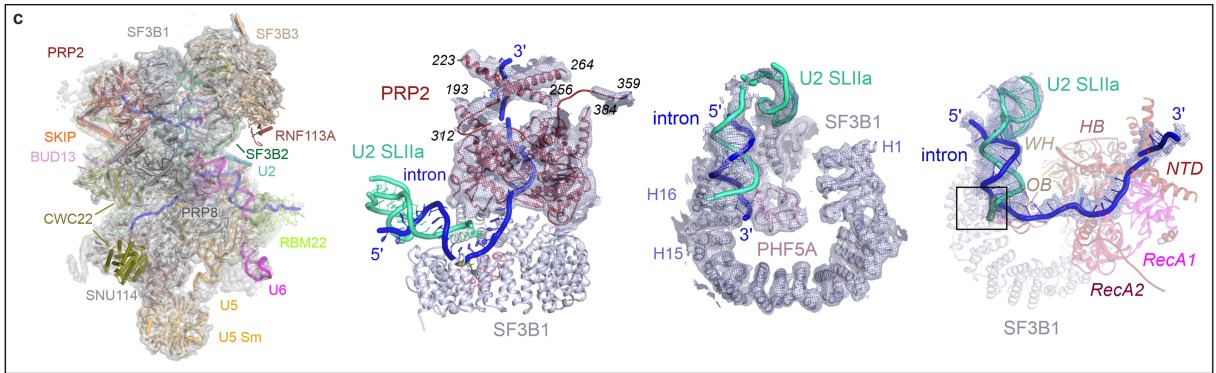

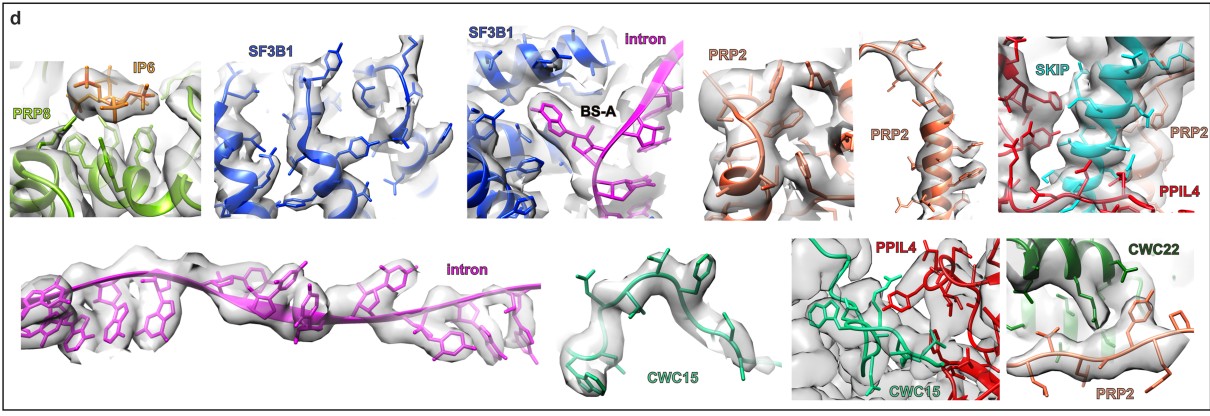

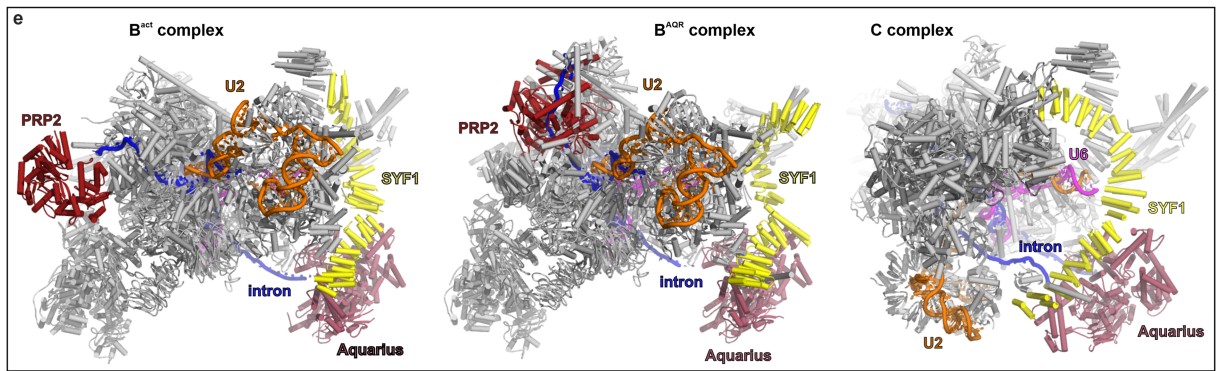

**Extended Data Fig. 3 | Cryo-EM maps and structure of the B^AQR spliceosome.**
**a**, Overall maps of state A and B complexes depicted together with the final model. **b**, Local resolution of the state A maps, estimated in RELION and visualized in ChimeraX. **c**, Cryo-EM density snapshots. Various subunits are colored and labeled. **d**, Selected snapshots of modeled B^AQR subunits are depicted together with the unsharpened map of the core region of the complex (map M2).

**e**, Structural comparison between B^act, B^AQR, and C complexes. PRP2, Aquarius, SYF1, the U2 snRNA, and the pre-mRNA substrate are colored and labeled. Note the large-scale repositioning of the PRP2 RNA helicase during the conversion of B^act to B^AQR, and of the helicase Aquarius during the transition of B^act/B^AQR complexes to the C-stage spliceosome.

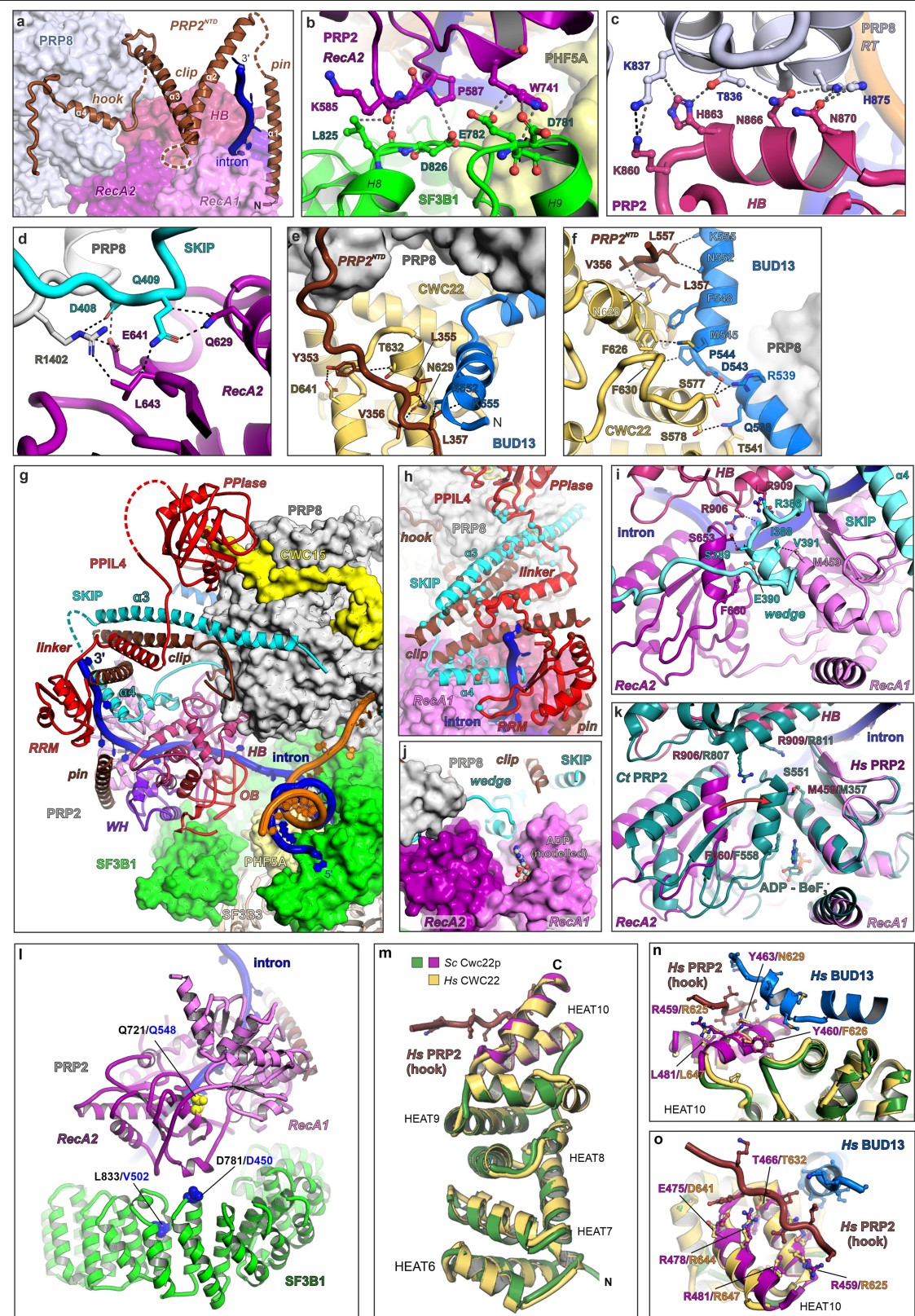

**Extended Data Fig. 4** | See next page for caption.

**Extended Data Fig. 4 | Functionally-important contacts in the B$^{AQR}$ spliceosome. a–f**, Interactions between PRP2, PRP8, SF3B1, CWC22 and SKIP. Interacting distances corresponding to polar and hydrophobic contacts are indicated as dashed lines. **g**, Overview of PRP2 and the composite molecular brake. PRP2's conserved domains are color-coded. HB – helix-bundle; OB – oligonucleotide/oligosaccharide- binding fold; WH – winged-helix. **h**, Interactions between PPIL4, SKIP, and PRP2. Residues involved in contacts are depicted as spheres and shown in the same color as their interacting partner. **i**, Interactions between the wedge element of SKIP and PRP2$^{core}$. **j**, The wedge element of SKIP appears to lock the RecA domains of PRP2. The ADP molecule is modeled by superposition with *Ct* PRP2 (PDB 6zm2) and is shown for the sake of orientation. **k**, Superposition between human PRP2 in the open conformation (observed in B$^{AQR}$) and *Ct* PRP2 in the closed conformation (colored in teal) (PDB 6zm2). Equivalent residues of PRP2 interacting with SKIP in B$^{AQR}$ are shown for *Ct* PRP2. Structures from panels i and k are depicted in the same orientation.

The red arrow indicates the movement of the RecA2-like domain during the helicase transition from the open to the closed conformation. **l**, Genetic interactions from yeast mapped on the structure of human B$^{AQR}$. The cold-sensitive allele of Prp2p$^{Q548N}$ genetically interacts with the D450G and V502F substitutions of Hsh155p. Residues involved in genetic interactions in yeast (blue) are depicted as spheres and mapped on the B$^{AQR}$ model. **m-o**, Structural superposition of the human (*Hs*) and budding yeast (*Sc*) MA3 domain of CWC22, composed from HEAT repeats. The 10$^{th}$ HEAT repeat (residues 454–491) of Cwc22p is required for the productive function of yeast Prp2p and interacts with the "hook" motif of PRP2$^{NTD}$ and BUD13 in B$^{AQR}$. This suggests that Prp2p, Cwc22p, and Bud13p may interact in a similar fashion in budding yeast spliceosomes, thus explaining the functional connection between Prp2p and Cwc22p. Note that human CWC22 residues that interact with PRP2 and BUD13 are conserved in budding yeast Cwc22p (shown in n and o). The protein subunits, U2 snRNA and the pre-mRNA substrate are colored and labeled.

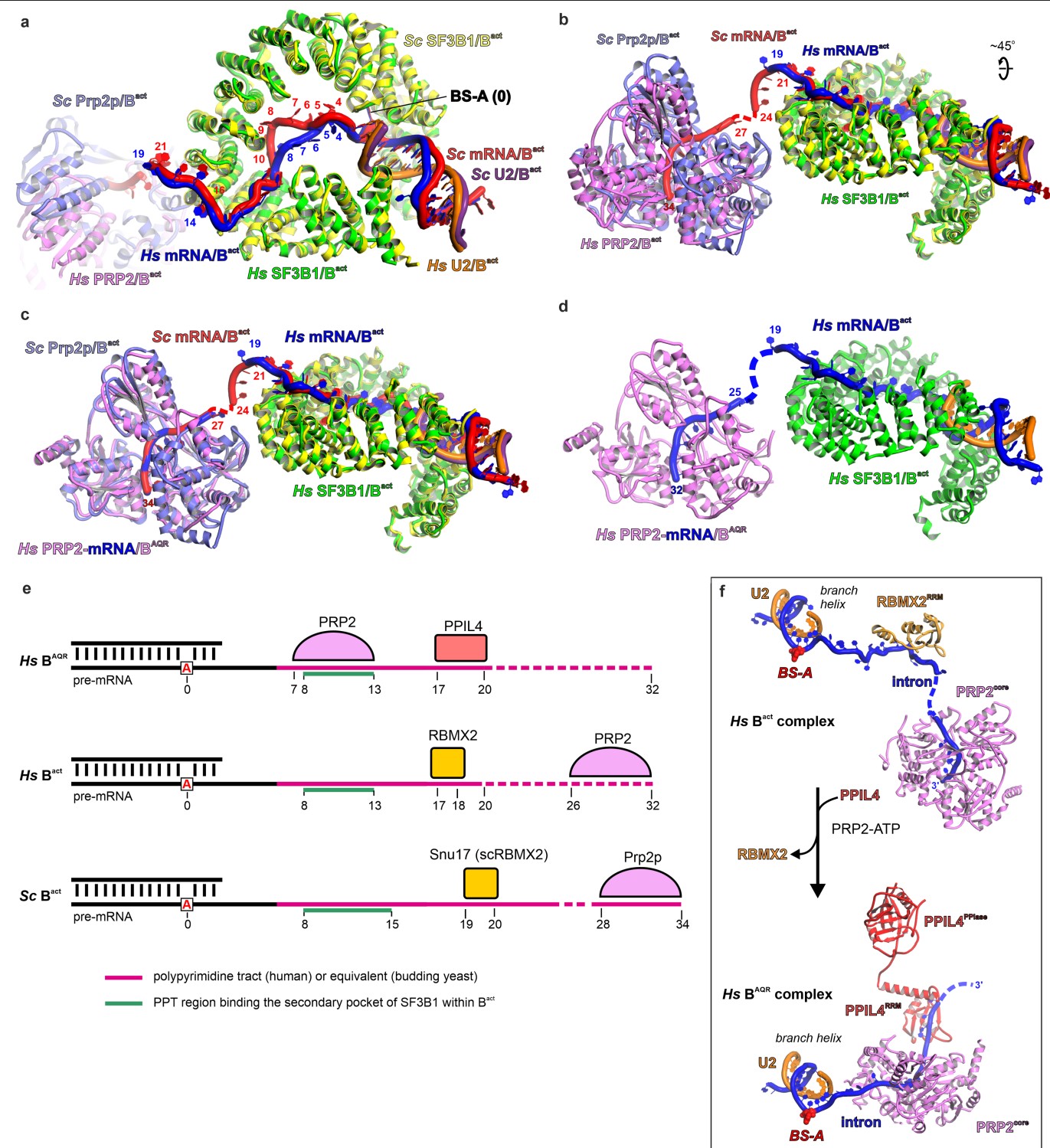

**Extended Data Fig. 5 | Estimation of the number of nucleotides translocated by PRP2 during the B^act-to-B^AQR transition. a,b,** Superposition between budding yeast (PDB 7DCO) and human (PDB 5Z56) B^act complexes over equivalent residues from PHF5A. For clarity's sake, only SF3B1, PRP2, the intron and U2 snRNA are depicted. All subunits are labeled accordingly. **c,** PRP2:RNA from human B^AQR, as the only available structure of the human counterpart, was superimposed onto the Prp2p helicase from budding yeast B^act and shown in the same orientation as in b. Note that the conformation of the RNA strand bound by human PRP2 or budding yeast Prp2p is virtually unchanged. **d,** Assignment of nucleotides bound by human PRP2 in B^act based on the superposition. **e,** PRP2 translocates about 19 nucleotides towards the branch helix during the B^act-to-B^AQR transition. PRP2, RBMX2 and PPIL4 positions on the intron in different spliceosomal complexes are depicted. **f,** PRP2 translocation results in a substantial change in the intron's conformation, dissociation of RBMX2 and recruitment of PPIL4 to the intron.

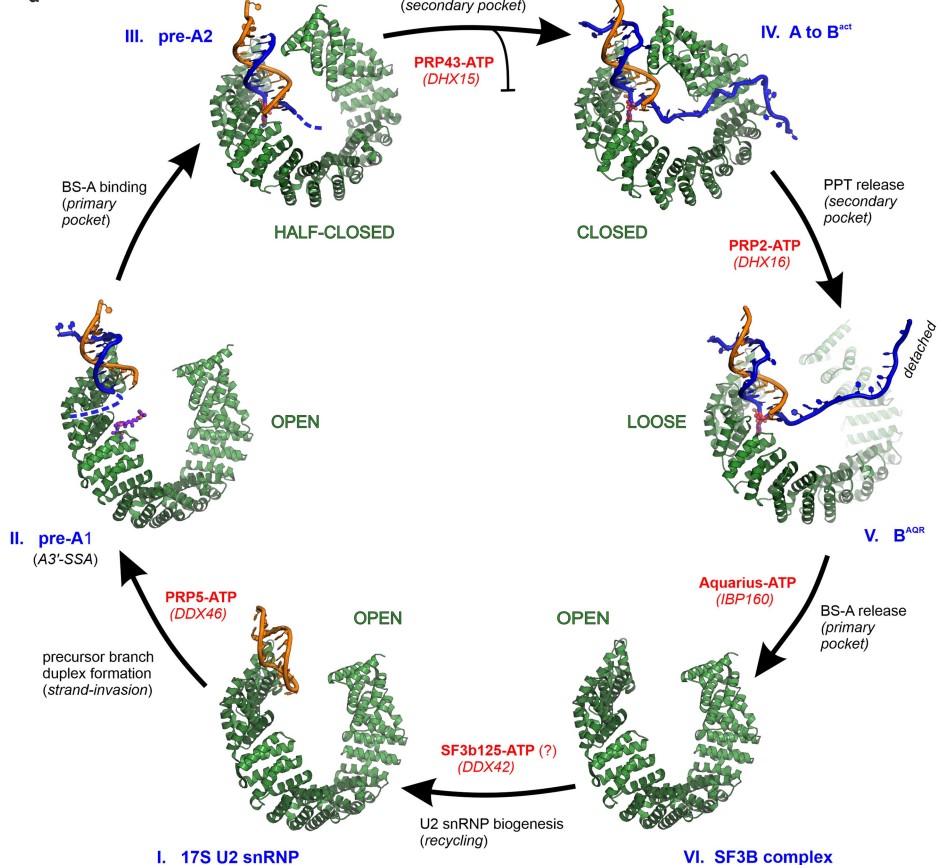

**Extended Data Fig. 6** | See next page for caption.

**Extended Data Fig. 6 | Conformational transitions of SF3B1. a**, Destabilization of SF3B6, de-structuring of SF3B1[NTD], and the reorientation of PRP8's EN (endonuclease-like, PRP8[EN]) and RH (RNase H-like, PRP8[RH]) domains upon PRP2 translocation. PRP2 is located on the plane above SF3B1[HEAT] and is not shown for clarity's sake. The reactive BS-A, the guanosine of the 5'SS, and the catalytic metal ions are shown as spheres and labeled. **b**, Close-up view of SF3B1, the intron, and the U2 snRNA from B[AQR] (color-coded) and B[act] (grey and black) after superposition of SF3B1's equivalent residues. **c**, The binding pocket of the BS-A (primary pocket) is virtually unchanged in B[act] and B[AQR]. **d**, Cycle of SF3B1 transitions in splicing. The conformational transitions of SF3B1 are depicted based on cryo-EM structures and biochemical analyses. Here we refer to the intermediates II and III as pre-A1 and pre-A2 for clarity. The key spliceosome complexes representative for the SF3B1 intermediates shown here are: 17S U2 snRNP[48,60], pre-A1 (A-like cross-exon complex bound by spliceostatin A[47]), pre-A2 (ref. 48), A-to-B[act] (reviewed in refs. 8,10,61), B[AQR] (this work), the SF3B complex[46,62]. Helicases that facilitate the transitions are shown in red. The helicase DHX15, colored magenta, mediates the disassembly of kinetically-slowed complexes (e.g. formed on suboptimal introns, weak splice sites and PPTs, multiple branch sites or cryptic sites[63–65]).

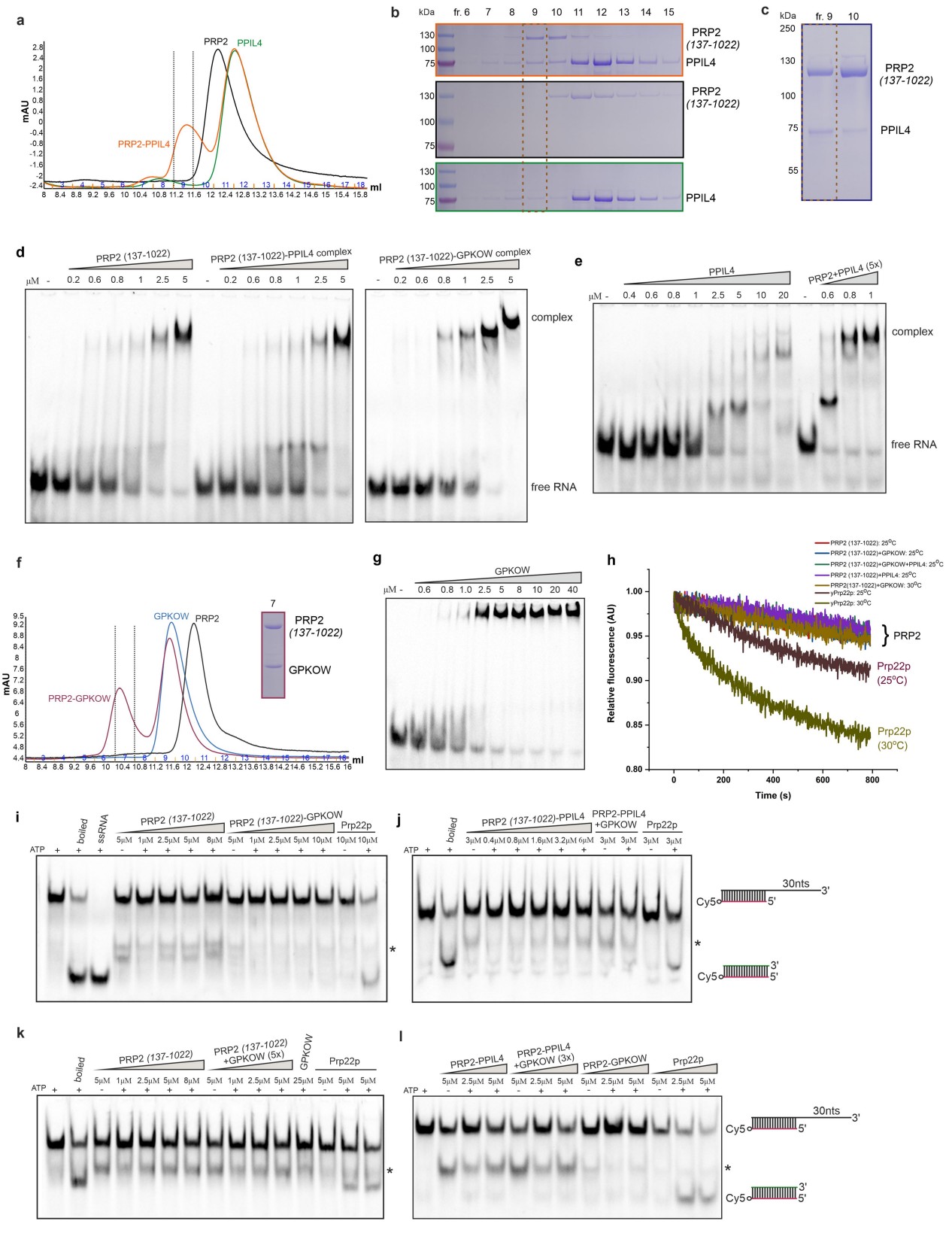

**Extended Data Fig. 7** | See next page for caption.

**Extended Data Fig. 7 | Biochemical analysis of human PRP2. a**, Size-exclusion chromatography (SEC) profiles of PRP2 (137-1022), PPIL4 and the mixture of the two proteins shows that PRP2 (137-1022) forms a stable complex with PPIL4, in an RNA-independent manner. PRP2 (137-1022) comprises both the modeled PRP2$^{NTD}$ (*i.e.*, the pin, clip, and hook elements) and the helicase core. **b**, SDS-PAGE gels corresponding to the fractions from **a**. The *in vitro* reconstitution experiments were repeated two times, using two independent preparations of PRP2 (137-1022) and PPIL4. **c**, SDS-PAGE gel of SEC fractions showing the reconstitution of a stable PRP2-PPIL4 complex *in vivo* by co-expression in insect cells. PRP2 (137-1022) and PPIL4 were co-expressed in Sf9 insect cells from individual baculoviruses. The complex was captured by Strep-Tactin affinity followed by SEC. The preparation was performed twice with similar results **d**, The RNA-binding activity of human PRP2 in the presence of GPKOW and PPIL4. The ability of PRP2 (137–1022) and of the purified PRP2 (137–1022)-PPIL4 and PRP2 (137-1022)-GPKOW complexes to bind a Cyanine 5 (Cy5)-labeled RNA substrate was evidenced by EMSA. The RNA substrate comprised an RNA duplex, followed by 30 nucleotides 3' single-stranded overhang, mimicking PRP2's spliceosome substrate observed in B$^{AQR}$. The free RNA substrate was separated from PRP2-bound (or cofactor-bound) species on a 5% polyacrylamide native gel. The EMSA gels were imaged at the Cy5 excitation peak. The assays were repeated two times. **e**, The RNA binding activity of human PPIL4 in the absence and presence of PRP2 (137–1022). Compared to the EMSA shown in **d**, PPIL4 was added in a 5-fold excess over PRP2 (137–1022) and PRP2's final assay concentrations are indicated above the last three gels lanes. Note the apparent increase in PRP2's RNA affinity in the presence of PPIL4. The EMSAs were repeated three times. **f**, SEC profiles showing the formation of a stable complex between PRP2 (137–1022) and GPKOW, in an RNA-independent manner. The SDS-PAGE corresponding to the complex purified by SEC is shown. **g**, The RNA-binding activity of human GPKOW assessed by an EMSA. The same RNA

substrate was used as in **d** and the assay was repeated four times. **h**, The helicase activity of human PRP2 was investigated using a fluorescence-based assay. Compared to the gel-based helicase assays shown in **i-l**, the fluorescence-based assay employs a dual-labeled helicase substrate. Displacement of the labeled strand by the helicase leads to the formation of an intramolecular hairpin, which brings in proximity the Cy5 fluorophore and its spectrally overlapping quencher (BHQ-2). The decrease in substrate's fluorescence upon unwinding is monitored as a function of time. Several representative fluorescence traces of PRP2, recorded under different experimental conditions and in the presence/absence of helicase cofactors, are shown together with the unwinding curves of Prp22p, used as a positive control. **i**, The helicase/unwinding activity of human PRP2 and of the PRP2-GPKOW complex. To assess the ability of PRP2 (137–1022) and of the purified, *in vitro* reconstituted PRP2 (137–1022)-GPKOW complex to unwind RNA-RNA duplexes, the purified protein samples were mixed with the Cy5-labeled helicase substrate (depicted on the right with the labeled strand colored in red) in the presence (or absence) of ATP and of a competitor DNA (green). Following a 1-hour incubation, the samples were analyzed on a native 14% polyacrylamide gel and imaged by in-gel fluorescence. Budding yeast Prp22p was used as a positive control. All gel-based helicase assays were repeated at least two times. **j**, The helicase activity of the *in vivo* reconstituted PRP2 (137–1022)-PPIL4 complex, in the absence or presence of GPKOW. **k**, The helicase activity of human PRP2 (137–1022) in the presence or absence of its cofactor GPKOW. Compared to the assay shown in **i**, the GPKOW cofactor was added to the purified helicase in a 5-fold excess. The concentrations indicated above the native gel represent the final assay concentrations of PRP2 (137–1022) or Prp22p. The RNA bands labeled with an asterisk represent, most likely, degradation products. **l**, Comparative helicase activities of the PRP2 (137–1022)-PPIL4 and PRP2 (137–1022)-GPKOW complexes. For gel source data, see Supplementary Figs. 2 and 3.

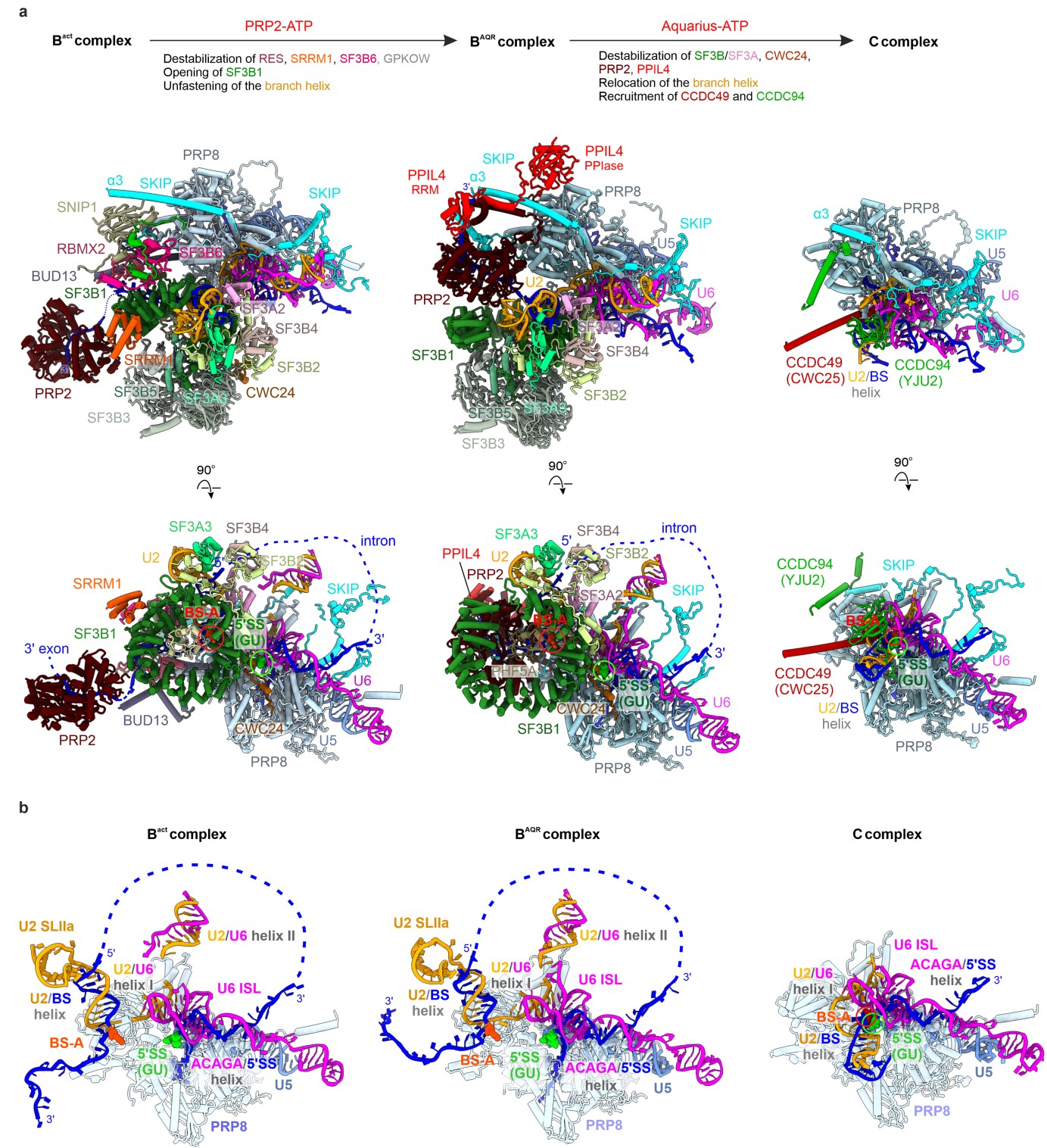

**Extended Data Fig. 8 | Remodeling events at the transition from B[act] to B[AQR] and then to C complexes. a**, During the B[act] (PDB 5Z57) to the B[AQR] transition, translocation of PRP2 in a 3'-to-5' direction results in the destabilization of the RES complex (RBMX2, BUD13, SNIP1). In addition, SRRM1 and SF3B6/p14 are no longer observed in B[AQR] due to PRP2-induced SF3B1 opening. During the remodeling of B[AQR] and transition to the C complex (PDB 5yzg, PDB 6zym), the remaining SF3B/SF3A subunits and CWC24 are released from the branch helix and the 5'SS. The branch helix then moves to the catalytic center, bringing the BS-A and the 5'SS GU nucleotides in proximity for the branching reaction. The different human spliceosome states were structurally aligned by using the

PRP8 subunit as a reference and are depicted in two different orientations. The spliceosome subunits are color-coded and shown in cartoon representation. Spliceosome subunits not undergoing significant rearrangement were omitted for the sake of simplicity. **b**, Repositioning of the branch helix (U2/BS) during the transition from B[act] to B[AQR] and then to C complexes. The different spliceosome states, B[act] (PDB 5Z57), B[AQR] (this work), and C complex (PDB 5yzg, PDB 6zym), were aligned using the PRP8 subunit as a reference. All protein subunits, except PRP8, were omitted and the RNA moieties were color-coded. The reactive BS-A and the 5'SS GU nucleotides are shown as spheres and colored red and light green, respectively.

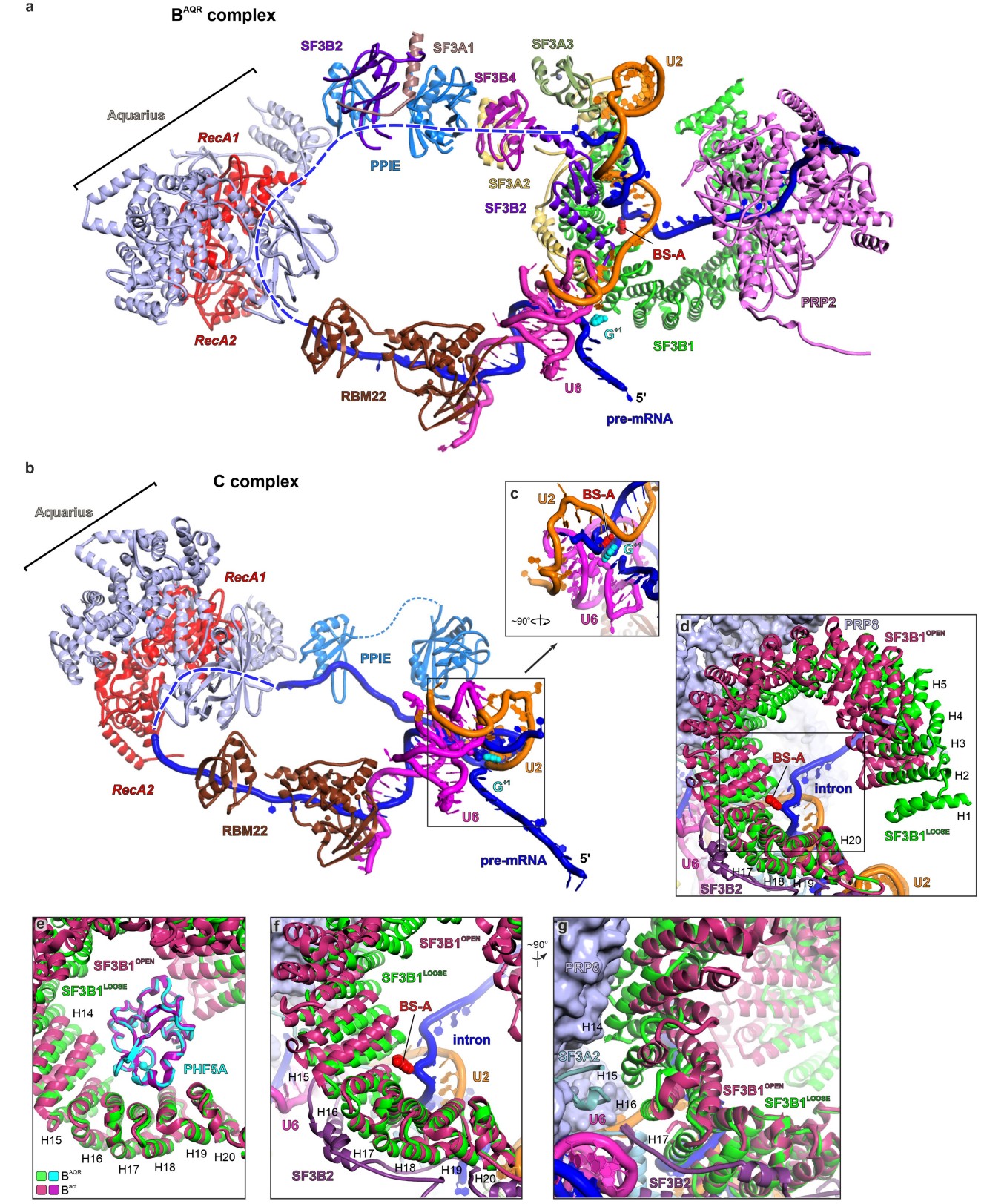

**Extended Data Fig. 9** | See next page for caption.

**Extended Data Fig. 9 | Aquarius promotes the transfer of the BS-A to the catalytic center during B$^{AQR}$-to-C complex transition, likely by inducing the "loose" to "open" conformational transition of SF3B1. a**, Lines of structural communication between Aquarius and the branch duplex in B$^{AQR}$. Aquarius and PRP2 are in diametrically-opposed locations of the B$^{AQR}$ spliceosome. A continuous bridge of proteins is present between Aquarius and the SF3B1:branch duplex. Most of these proteins are SF3A/B subunits and PPIE. Another bridge that primarily involves the RBM22 protein is between Aquarius and U6 catalytic core. The intron region not visible in the density is dashed. Aquarius is depicted in red (RecA-like domains) and light blue (accessory domains). The first nucleotide of the intron (G$^{+1}$) is positioned in the catalytic center. Other subunits of the spliceosome, including SYF1 and ISY1, are not shown for the sake of clarity **b**, Subunits of the C complex (pdb 5yzg) are shown in the same orientation as in **a**. The U6 snRNA was used as a reference for the superposition between the C and B$^{AQR}$ complexes. Note that PRP2 and the SF3A/B complex have dissociated and the BS-A has been relocated to the catalytic centre (see also **c**, below). RBM22's orientation has remained virtually the same in B$^{AQR}$

and C complexes. The subunits are colored as in a and labeled. The B$^{AQR}$ and C complexes were superimposed over PRP8 (not shown) and U6 snRNA **c**, The BS-A juxtaposed to the first nucleotide of the intron in the catalytic center of the C complex. The catalytic metal ions are shown as magenta spheres. **d**, Superposition between the "open" (observed in the apo SF3B complex, PDB 5IFE) and the "loose" states (observed in B$^{AQR}$) of SF3B1. Except for SF3B1$^{OPEN}$, all shown subunits belong to the B$^{AQR}$ complex. **e**, B$^{AQR}$ and the SF3B complex in its apo form (PDB 5IFE) were superimposed over equivalent residues from PHF5A (not shown in the figures d, f and g for clarity's sake). Note that the HEAT repeats H16-H20 of SF3B1 adopt virtually identical conformations in the "loose" and "open" states, while the other helical repeats are substantially reorganized. **f–g**, The BS-A's release from the primary pocket of SF3B1 likely induces rearrangement of the HEAT repeats upon the "loose" to "open" conformational transition. Consequently, the contacts between HEAT repeats and PRP8, and those between SF3B2 (attached to the HEAT repeats) and the U6 snRNA might get disrupted, causing the complete dissociation of the SF3A/B complexes from the spliceosome.

**Extended Data Table 1 | Data collection, model refinement and validation**

| | Map M2 (State A, overall) | | Map M2 (State A, core) | Map M4 (State B, overall) | Map M5 (State A, U2 3'/IBC) |
|---|---|---|---|---|---|
| **Data collection and processing** | 0° | 25° | | | |
| Electron gun | XFEG | XFEG | | | |
| Detector | K2 | K2 | | | |
| Magnification | 130000 | 130000 | | | |
| Energy filter slit width (eV) | 30.0 | 30.0 | | | |
| Voltage (kV) | 300.0 | 300.0 | | | |
| Dose rate($e^-$/A$^2$/s) | 5.04 | 5.06 | | | |
| Electron exposure on sample ($e^-$/A$^2$) | 45.38 | 45.55 | | | |
| Target defocus range (μm) | 1.0-2.5 | 1.0-2.5 | | | |
| Calibrated pixel size (Å) | 1.05 | 1.05 | | | |
| Symmetry imposed | C1 | C1 | | | |
| Collected movies (no.) | 5229 | 4784 | | | |
| Initial particle images (no.) | 734691 | | | | |
| Final particle images (no.) | 146157 | | 146157 | 12395 | 27719 |
| Map resolution at FSC=0.143 (Å) | 3.1 | | 3.1 | 5.9 | 3.8 |
| **Refinement** | | | | | |
| Initial model used (PDB code) | | | 6FF4, 6FF7, 5Z57 | | |
| Model resolution (Å) | | | 3.2 | | |
| Model resolution at FSC=0.5 (Å) | | | 3.4 | | |
| Map sharpening B factor (Å$^2$) | | | -110.11 | | |
| Model composition | | | | | |
| Non-hydrogen atoms | | | 73542 | | |
| Protein residues | | | 9759 | | |
| RNA nucleotides | | | 343 | | |
| Ligands | | | 1x GTP, 1x IHP | | |
| Waters | | | - | | |
| Ions | | | 12x Zn$^{2+}$, 7x Mg$^{2+}$ | | |
| *B* factors (Å$^2$) | | | | | |
| Protein | | | 175.77 | | |
| Nucleotide | | | 220.7 | | |
| Ligand | | | 126.02 | | |
| Water | | | - | | |
| R.m.s. deviations | | | | | |
| Bond lengths (Å) | | | 0.004 | | |
| Bond angles (°) | | | 0.848 | | |
| **Validation** | | | | | |
| Molprobity score | | | 1.49 | | |
| Clashscore | | | 3.89 | | |
| Rotamer outliers (%) | | | 0.04 | | |
| Cβ outliers (%) | | | 0 | | |
| CaBLAM outliers (%) | | | 3.04 | | |
| Ramachandran plot | | | | | |
| Favored (%) | | | 95.43 | | |
| Allowed (%) | | | 4.32 | | |
| Disallowed (%) | | | 0.25 | | |

# Reporting Summary

## Statistics

For all statistical analyses, confirm that the following items are present in the figure legend, table legend, main text, or Methods section.

| n/a | Confirmed | |
|---|---|---|
| ☐ | ☒ | The exact sample size (*n*) for each experimental group/condition, given as a discrete number and unit of measurement |
| ☒ | ☐ | A statement on whether measurements were taken from distinct samples or whether the same sample was measured repeatedly |
| ☒ | ☐ | The statistical test(s) used AND whether they are one- or two-sided<br>*Only common tests should be described solely by name; describe more complex techniques in the Methods section.* |
| ☒ | ☐ | A description of all covariates tested |
| ☒ | ☐ | A description of any assumptions or corrections, such as tests of normality and adjustment for multiple comparisons |
| ☒ | ☐ | A full description of the statistical parameters including central tendency (e.g. means) or other basic estimates (e.g. regression coefficient) AND variation (e.g. standard deviation) or associated estimates of uncertainty (e.g. confidence intervals) |
| ☒ | ☐ | For null hypothesis testing, the test statistic (e.g. *F*, *t*, *r*) with confidence intervals, effect sizes, degrees of freedom and *P* value noted<br>*Give P values as exact values whenever suitable.* |
| ☒ | ☐ | For Bayesian analysis, information on the choice of priors and Markov chain Monte Carlo settings |
| ☒ | ☐ | For hierarchical and complex designs, identification of the appropriate level for tests and full reporting of outcomes |
| ☒ | ☐ | Estimates of effect sizes (e.g. Cohen's *d*, Pearson's *r*), indicating how they were calculated |

*Our web collection on statistics for biologists contains articles on many of the points above.*

## Software and code

Policy information about availability of computer code

| Data collection | The cryo-EM data were collected using EPU on a Titan Krios G2 electron microscope operated at 300 keV in EFTEM mode and equipped with a Quantum LS 967 energy filter (Gatan) and a K2 Summit direct electron detector (Gatan). |
|---|---|
| Data analysis | CCP4 (v. 7.0 .045), CCPEM (v. 1.4.1), Phenix (v. 1.19-4092), Coot (v. 0.9.6), UCSF Chimera (v. 1.14, v. 1.16), ChimeraX (v. 1.3, v. 1.5), Warp, cryoSPARC v0.65, RELION 3.0, RELION 3.1-beta, Pymol (v. 2.4.1), GraphPad Prism 9 (v. 9.5.0), OriginPro 2020 (v. 9.7), DeepEMhancer, Alphafold 2, ColabFold (v.1.5) |

For manuscripts utilizing custom algorithms or software that are central to the research but not yet described in published literature, software must be made available to editors and reviewers. We strongly encourage code deposition in a community repository (e.g. GitHub). See the Nature Portfolio guidelines for submitting code & software for further information.

## Data

Policy information about availability of data

All manuscripts must include a data availability statement. This statement should provide the following information, where applicable:
- Accession codes, unique identifiers, or web links for publicly available datasets
- A description of any restrictions on data availability
- For clinical datasets or third party data, please ensure that the statement adheres to our policy

The structural models and cryo-EM maps of the BAQR spliceosome (the core region and the state B holocomplex) were deposited in the Protein Data Bank (PDB accession codes 7QTT and 8CH6) and Electron Microscopy Data Bank (accession codes EMD-14146 amd EMD-16658), respectively.

## Human research participants

Policy information about studies involving human research participants and Sex and Gender in Research.

| | |
|---|---|
| Reporting on sex and gender | *Use the terms sex (biological attribute) and gender (shaped by social and cultural circumstances) carefully in order to avoid confusing both terms. Indicate if findings apply to only one sex or gender; describe whether sex and gender were considered in study design whether sex and/or gender was determined based on self-reporting or assigned and methods used. Provide in the source data disaggregated sex and gender data where this information has been collected, and consent has been obtained for sharing of individual-level data; provide overall numbers in this Reporting Summary. Please state if this information has not been collected. Report sex- and gender-based analyses where performed, justify reasons for lack of sex- and gender-based analysis.* |
| Population characteristics | *Describe the covariate-relevant population characteristics of the human research participants (e.g. age, genotypic information, past and current diagnosis and treatment categories). If you filled out the behavioural & social sciences study design questions and have nothing to add here, write "See above."* |
| Recruitment | *Describe how participants were recruited. Outline any potential self-selection bias or other biases that may be present and how these are likely to impact results.* |
| Ethics oversight | *Identify the organization(s) that approved the study protocol.* |

Note that full information on the approval of the study protocol must also be provided in the manuscript.

# Field-specific reporting

Please select the one below that is the best fit for your research. If you are not sure, read the appropriate sections before making your selection.

☒ Life sciences          ☐ Behavioural & social sciences          ☐ Ecological, evolutionary & environmental sciences

For a reference copy of the document with all sections, see nature.com/documents/nr-reporting-summary-flat.pdf

# Life sciences study design

All studies must disclose on these points even when the disclosure is negative.

| | |
|---|---|
| Sample size | Sample-size calculation was not performed. All biological/biochemical experiments were replicated two or more times. Two cryo-EM datasets of the BAQR spliceosome were collected at 0 and 25 tilt angles. |
| Data exclusions | Contaminated and low-resolution cryo-EM micrographs were excluded from the analysis with Warp. |
| Replication | All attempts at replication were successful. |
| Randomization | No randomization was required. |
| Blinding | Blinding is not applicable for the methods used in this study. |

# Reporting for specific materials, systems and methods

We require information from authors about some types of materials, experimental systems and methods used in many studies. Here, indicate whether each material, system or method listed is relevant to your study. If you are not sure if a list item applies to your research, read the appropriate section before selecting a response.

## Materials & experimental systems

| n/a | Involved in the study |
|---|---|
| ☒ | ☐ Antibodies |
| ☐ | ☒ Eukaryotic cell lines |
| ☒ | ☐ Palaeontology and archaeology |
| ☒ | ☐ Animals and other organisms |
| ☒ | ☐ Clinical data |
| ☒ | ☐ Dual use research of concern |

## Methods

| n/a | Involved in the study |
|---|---|
| ☒ | ☐ ChIP-seq |
| ☒ | ☐ Flow cytometry |
| ☒ | ☐ MRI-based neuroimaging |

# Eukaryotic cell lines

Policy information about cell lines and Sex and Gender in Research

| | |
|---|---|
| Cell line source(s) | High Five (Trichoplusia Ni, BTI-Tn-5B1-4) insect cells were cultured in ESF921 media and obtained from ThermoFisher Scientific, Cat#B85502 or Expression Systems, Cat#94-002F<br>Sf9 (Spodoptera frugiperda) insect cells were cultured in Sf-900 III SFM and obtained from ThermoFisher Scientific, Cat#11496015 and Cat#12659017.<br>HeLa S3 cells, used for nuclear extract preparation, were obtained from Helmoltz Center for Infection Research, Brunswick. |
| Authentication | Insect cell lines were not authenticated. |
| Mycoplasma contamination | Insect cell lines were not tested for Mycoplasma contamination. HeLa S3 cells were tested for Mycoplasma contamination. |
| Commonly misidentified lines<br>(See ICLAC register) | No commonly misidentified cell lines were used. |

