## [Peer Review File · Nature]

Manuscript Title: Structural basis of catalytic activation in human splicing

Reviewer Comments & Author Rebuttals

Reviewer Reports on the Initial Version:

Referees' comments:

Referee #1:

In this manuscript, the authors present the cryo-EM structure of a human spliceosome intermediate stalled at the catalytic activation step. They used a dominant-negative Aquarius mutant to block the splicing reaction, and arrested the spliceosome halfway through catalytic activation. The structure, which they call BAQR, reveals several interesting features distinct from that of Bact, including the localization of PRP2 near the spliceosome core and the open conformation of SF3B1. The N-terminal disordered domain of PRP2, which is visualized only in BAQR, interacts with spliceosomal components PPIL4 and SKIP, and the authors propose that it plays a regulatory role for structural changes. BAQR represents a spliceosome intermediate in the transition from Bact to B*. A model is proposed for how PRP2 may mediate structural changes of the spliceosome from Bact to BAQR.

Although the release of U1 and U4 leads to the activation of the spliceosome to form the RNA catalytic core, the branch helix is still 50 Å away from the catalytic center, hindered by SF3B. Biochemical and proteomic studies have revealed that PRP2 plays a crucial role in removing SF3 complexes to allow the positioning of the branch site at the catalytic center. How PRP2 functions in remodeling of the spliceosome has been studied mostly in yeast, which does not have an Aquarius homolog. The use by the authors of the Aquarius mutant to arrest the spliceosome intermediate at catalytic activation represents an excellent approach for identifying the intermediate form in the human system and delineating detailed mechanisms for this process. The results demonstrate that PRP2 translocates from the periphery of the spliceosome toward the spliceosome core to displace components binding on the branch site downstream region. These results also argue against the pulling model for the action of DEAH RNA helicases. They also echo findings from previous studies in yeast, showing that PRP2 binds to the branch site downstream region, and then translocates in a 3' to 5' direction toward the branch site to displace SF3 complexes (Liu, 2012). The structure presented in this work uncovers details of the interactions among components on BAQR and, by comparing the structure of BAQR with those of Bact and B*, reveals how PRP2 and Aquarius may function in remodeling of the human spliceosome. Moreover, it reveals an interesting two-step process for catalytic activation of the human spliceosome. Overall, this paper represents a significant advance in

our understanding of the mechanism of catalytic activation, and should be of great interest to people in the splicing field.

However, the paper falls short in that the Discussion is too superficial, simply representing a summary of the results. The authors should compare their findings with previous work on yeast to explore mechanistic similarities and differences. Yeast does not have an Aquarius homolog, so PRP2 is responsible for the entire process. PRP2 has been extensively studied in yeast. A model for how PRP2 may function in catalytic activation has also been proposed. These works should be cited and discussed. It would also be interesting to discuss if the spliceosome stalls the same way in the absence of Aquarius as when a dominant-negative mutant of Aquarius is present, and how Aquarius might work in the second phase of the process based on the structural information for BAQR.

Comments:

1. The diagram in Fig. 1b is misleading. It gives the impression that RES is removed from the spliceosome during the transition from Bact to BAQR, yet RES is only removed away from the pre mRNA without being released from the spliceosome.
2. Representations of pink, green, and broken lines should be indicated in Fig. 3a.
3. P.2, line 39: It is not clear to me how the authors determined that PRP2 translocates 18 nucleotides from the periphery towards the spliceosome core. Is the region of the pre-mRNA that PRP2 binds to on the Bact structure clearly visualized so that the distance of the translocation can be measured?
4. P.3, line 53: The description of "The first reaction.....within the C complex spliceosome" is imprecise. The C complex is the product of the first reaction, and is a stable form of the complex that can be isolated and analyzed.
5. P.3, line 68: The authors appear unaware of and have failed to cite previous studies on yeast related to how PRP2 may drive catalytic activation (Liu, 2012). Two other DEAH RNA helicases, PRP16 and PRP22, have also been characterized in yeast, and a similar mechanism for remodeling the spliceosome in the second catalytic step and for the release of mature mRNA has been proposed (Chung, 2019; Schwer, 2008). The authors should discuss if this mechanism is general for DEAH RNA helicases.
6. P.5, line 120: How can one explain the observation that PRP2 on Sc Bact exhibits the conformation corresponding to a post-ATP hydrolytic state prior to PRP2 activity?
7. P.10, line 238: PRP2....translocated to "an upstream" position.

8. P.12: "SPP2" instead of "Spp2" for consistency.
9. P.12: This result is consistent with a previous report that the NTD of Sc PRP2 is dispensable (Edwalds-Gilbert, 2004), which should be included in the Discussion.
10. The model in Fig. 5b should be explained in more detail, in particular, how the hypothetical intermediate is generated. It would also be more appropriate to place the summary of the respective results in the legend of Fig. 5b than in the Discussion.

Referee #2:

This manuscript provides critical new concepts to our understanding of spliceosomal activation, i.e., the process by which, just prior to catalytic step one, the first-step nucleophile — the branch site adenosine complexed with U2 snRNA — is released from SF3b1 and moves 50A into the active site. The authors present a ~2.9Å cryo-EM structure of a human spliceosomal complex stalled by a mutation in the ATPase Aquarius. This represents a previously unknown intermediate in human spliceosome assembly/activation, which is not present in the yeast system (which does not have an Aquarius ortholog). The manuscript provides key new insights into the function of both PRP2 and Aquarius ATPases in spliceosome activation.

Lines 130-131, "Whether DEAH-box helicases, including PRP2, pull their substrates from a distance or by stepwise translocation along the RNA strand remained unclear.": This is a big point, and I want to reiterate it and say that I entirely agree with the authors' assertion here. I don't think that we have known what any spliceosomal ATPase really does, and this is a big step forward in understanding PRP2 function. By stalling the complex at the AQR step, the authors can, for the first time, infer exactly what PRP2 translocates on: "PRP2 advanced 18 nts [on the pre-mRNA] towards the spliceosome core during the Bact-to-BAQR transition". I think that this point and its concomitant effects on the structure are incredibly important and an appropriate topic of a paper in Nature. Overall, this is one of the few spliceosome structural papers in which we actually learn new, unanticipated features of the mechanism.

Specific Comments:

1. Line 83, "Following up on our previous work 6": This needs a bit more explanation.
2. The mechanism of translocation termination of PRP2 by "molecular brake elements" proposed by the authors is very interesting. It would be useful to (briefly) compare this to what is known about the termination of translocation of other helicases. For example, is it similar, or not, to the

mechanism by which MAGOH-Y14 inhibits eIF4AIII and locks it onto RNA within the exon junction complex (Ballot et al., NSMB, 2005; PMID 16170325)?

3. The action of the ATPases as molecular clamps seems a lot like what was proposed by Melissa Moore for DExH/D proteins (Shibuya et al., NSMB, 2004; PMID 15034551).

4. Line 377, Fig. 1, "Key subunits are color-coded": This could be clarified. I understand that the colors in the molecule structure schematic are a rough match with the colors for the components list, shown in a, lower. However, there are some inconsistencies. For example, PRP2 is shown in brown, but it is not color-matched in Fig. 1b, where PRP2 is inconsistently written in red. Why is the RES complex indicated in light blue, when the only protein component included, BUD13, is dark blue? Overall, there are too many colors and shades of colors here to be able to follow.

5. Lines 268-271: There is a high richness of new detail provided throughout. However, one statement, "The two helicases must coordinate to liberate the duplex from SF3B-RES constraints and hand it over to the catalytic center, while keeping it unaltered – a task that is best accomplished sequentially, by two molecular motors", leaves me wanting more info. Why is this "best accomplished" by two molecular motors? This is a critical difference between mammals and yeast, and I think it deserves to be explained further. Here, this is stated as a fact; as such, it needs further explanation. Alternatively, this could be stated as a conjecture that is, so far, without adequate explanation.

Line 378: "spliceosomes" should be "spliceosome".

Referee #3:

The manuscript by Schmitzova et al. presents the structure of a new spliceosome assembly intermediate, Baqr, captured via a dominant-negative mutation in the helicase Aquarius. Aquarius is the only SF1-family helicase that is involved in splicing and its function is not well understood, in particular, that it is absent in some model organisms (e.g., *S. cerevisiae*). The complex presented by the authors was stalled between Bact and B*/C states and provides important mechanistic insights into this transition.

Some of the key findings include:

1. Dissection of the Bact to B* transition into two steps, implying that PRP2 and Aquarius act in a sequential manner.

2. PRP2 was captured in a position suggesting that it translocates along the pre-mRNA in order to displace a specific subset of splicing, including the RES complex. It was previously proposed that PRP2 would remain attached to SF3B1 and act on the branch helix by pulling from a distance, but the exact mechanism of this remodeling remained elusive. The Baqr structure provides a conceptually new mechanism for this rearrangement.
3. PRP2 action alone is sufficient to unfasten the branch helix from the SF3b complex by re-opening SF3B1 and therefore priming it for further remodeling.
4. SKIP and PPIL4 associate with intron RNA exiting PRP2 and could act as a molecular brake to achieve precise control over the extent of PRP2 translocation.

Overall it is a beautiful work providing new and exciting insights into the mechanism of one of the key steps in spliceosome assembly. Technically, the biochemical and structural aspects of the work are of high quality. The manuscript is clearly written and easy to follow, with all findings accompanied by appropriate figures.

I would recommend publication of this work in Nature upon addressing some of my comments below.

Major points:

1. While I understand the authors' logic that PRP2 is required for the Bact to Baqr transition and that Aquarius is needed further down the line, I am not entirely convinced if one can simplify the whole process by stating that the two helicases act in a sequential manner. One could imagine a situation where the action of Aquarius could only be needed to enable PRP2 to continue its translocation until the branch helix is completely liberated. This could happen, for example, by translocating the remote parts of the intron upstream from the branch site to release some potential tensions in the system to allow PRP2 to continue its unwinding. In other words, it is possible that IBC-K829A mutation would keep IBC locked on one position of intron, preventing PRP2 from further translocation. What would speak in favor of such a hypothesis is the fact that Prp2 alone is capable of performing catalytic activation in some species (i.e., *S. cerevisiae*) without the need for Aquarius. The sequential action of the two helicases is an important point of the paper, and alternative scenarios should be at least thoroughly considered in the Discussion. Ideally, the authors should try to come up with a feasible experiment showing that PRP2 is not needed to complete the catalytic activation after initial translocation or prove that Aquarius is really needed to complete it.

2. The authors propose that SKIP and PPIL4 act as a molecular brake that stops PRP2 from unwinding branch duplex and suggest that both proteins act as negative regulators of PRP2 in a context-dependent manner. This is potentially a very important point as such a regulation mechanism is conceptually new and might also be used by other spliceosomal helicases. However, based on the structure alone, the causality remains ambiguous here (i.e., it is not clear to me if SKIP/PPIL4 could bind to PRPF2 and actively induce its opening/inhibition, or do they just associate with the repositioned PRP2, which is inhibited from translocating further by some other means). It would significantly strengthen the manuscript if the authors provided some experimental evidence that SKIP and PPIL4 can indeed influence the intrinsic activity of PRP2 (for example, by an *in vitro* helicase assay).

3. The key question that remains open is what is the role of Aquarius in the catalytic activation process? Although Aquarius was used by the authors to stall spliceosomes, the manuscript does not discuss its role in this process as all the analyses are focussed on PRP2. What is particularly puzzling is the fact that Aquarius binds upstream of the branch site, and given that it has 3'-5' RNA unwinding activity (De et al., NSMB, 2015), it is very difficult to imagine how it could exert any force on the branch helix, which still needs to be remodeled. By translocating with this polarity, it should create slack rather than tension between itself and the branch helix (see also point 1). I believe that the authors should comment on this aspect of their work. In particular, since the title "Structural basis of catalytic activation in human splicing" implies insights into the whole process of catalytic activation, but here only part of this process was analyzed.

Minor comments:

1. Line 68, "by a mechanism that remains largely hypothetical": I would suggest rephrasing to "by a mechanism that remains largely unknown".

2. Line 167, "Surprisingly, after releasing the RES complex from RNA and former protein contacts, the RES complex binds, via BUD13531-534, to a composite structure formed by the hook of PRP2NTD, the slightly relocated C-terminal MA3 domain of CWC22 and PRP8 (Fig. 3a and Extended Data Fig. 13b,c)": This is a bit misleading in suggesting that the entire RES complex is relocated, while it is only a small fragment of Bud13 that rebinds. This should be rephrased to avoid any confusion.

3. It would be helpful to see some examples of high-resolution fitting, as current cryo-EM density snapshots (Ext. Data Fig. 8) show mostly secondary structure elements and do not support high-resolution data claimed by the FSC. This is relevant given the highly anisotropic angular distribution of the 3D reconstruction, which could cause a mismatch between the resolution estimate and observed features.

4. There is a numbering problem with Ext. Data Figs. (12, 14, 14); 13 is missing. 5. In Table S2 some entries are color-coded, but the meaning of that is not explained.

Author Rebuttals to Initial Comments:

The referee's comments are in black and the point-by-point response in blue.

Referee #1:

In this manuscript, the authors present the cryo-EM structure of a human spliceosome intermediate stalled at the catalytic activation step. They used a dominant-negative Aquarius mutant to block the splicing reaction, and arrested the spliceosome halfway through catalytic activation. The structure, which they call BAQR, reveals several interesting features distinct from that of Bact, including the localization of PRP2 near the spliceosome core and the open conformation of SF3B1. The N-terminal disordered domain of PRP2, which is visualized only in BAQR, interacts with spliceosomal components PPIL4 and SKIP, and the authors propose that it plays a regulatory role for structural changes. BAQR represents a spliceosome intermediate in the transition from Bact to B*. A model is proposed for how PRP2 may mediate structural changes of the spliceosome from Bact to BAQR.

Although the release of U1 and U4 leads to activation of the spliceosome to form the RNA catalytic core, the branch helix is still 50 Å away from the catalytic center hindered by SF3B. Biochemical and proteomic studies have revealed that PRP2 plays a crucial role in removing SF3 complexes to allow the positioning of the branch site at the catalytic center. How PRP2 functions in remodeling of the spliceosome has been studied mostly in yeast, which does not have an Aquarius homolog. The use by the authors of the Aquarius mutant to arrest the spliceosome intermediate at catalytic activation represents an excellent approach for identifying the intermediate form in the human system and delineating detailed mechanisms for this process. The results demonstrate that PRP2 translocates from the periphery of the spliceosome toward the spliceosome core to displace components binding on the branch site downstream region. These results also argue against the pulling model for the action of

DEAH RNA helicases. They also echo findings from previous studies in yeast, showing that PRP2 binds to the branch site downstream region, and then translocates in a 3' to 5' direction toward the branch site to displace SF3 complexes (Liu, 2012). The structure presented in this work uncovers details of the interactions among components on BAQR and, by comparing the structure of BAQR with those of Bact and B*, reveals how PRP2 and Aquarius may function in remodeling of the human spliceosome. Moreover, it reveals an interesting two-step process for catalytic activation of the

human spliceosome. Overall, this paper represents a significant advance in our understanding of the mechanism of catalytic activation, and should be of great interest to people in the splicing field.

However, the paper falls short in that the Discussion is too superficial, simply representing a summary of the results. The authors should compare their findings with previous work on yeast to explore mechanistic similarities and differences. Yeast does not have an Aquarius homolog, so PRP2 is responsible for the entire process. PRP2 has been extensively studied in yeast. A model for how PRP2 may function in catalytic activation has also been proposed. These works should be cited and discussed.

We are very thankful for these suggestions. In the revised manuscript, we dedicate an entirely new section, with a subheading, to the catalytic activation in budding yeast and the parallels with human splicing (P13-P14). We have also included a model of catalytic activation in yeast, where we integrate our findings and the previously published structural, biochemical and genetics data (Fig. 5c). We tried to cite all essential references, especially from the labs of Soo Chen Cheng, Reinhard Lührmann, Ren Jang Lin, Yigong Shi and Aaron Hosking.

It would also be interesting to discuss if the spliceosome stalls the same way in the absence of Aquarius as when a dominant-negative mutant of Aquarius is present, and how Aquarius might work in the second phase of the process based on the structural information for B^{AQR}.

Besides acting as a helicase, Aquarius seems to possess a structural role in maintaining the stability of the pentameric IBC, securing the recruitment of the five components to the spliceosome (De et al., NSMB, 2015). The absence of Aquarius may therefore prevent proper incorporation of the other IBC components, including SYF1, at the B-to-B^{act} transition. Misincorporation of SYF1 (due to Aquarius's absence) might affect the microenvironment of the catalytic center, possibly preventing the formation of a functional B^{act} complex. Two main reasons suggest this scenario.

First, SYF1 interacts extensively with SYF3, whose N-terminal helical repeats (tetratricopeptides, TPRs) bind U6 within B^{act}, very close to the catalytic center, suggesting an essential role of the two proteins in assisting the formation of the RNA-based catalytic center. Second, the yeast homologs of SYF1 and SYF3 are essential for cell viability and belong to the Nineteen Complex (NTC). NTC in yeast is essential for the catalytic center formation (in particular for the association of U5 and U6 with the spliceosome after the dissociation of U4; Chan et al., Science 2003).

To conclude, we estimate that the absence of Aquarius might affect the B-to-B^{act} transition due to the scaffolding function of this helicase. In contrast, Aquarius^{K829A} stalls the spliceosome later, at the B^{AQR} state, as it affects the helicase function.

It would also be interesting to discuss .., how Aquarius might work in the second phase of the process based on the structural information for B^{AQR}.

In the revised manuscript, we discuss how Aquarius might work in the second phase of the catalytic activation (P12-P13); we amend the model from Fig. 5b with an additional intermediate and have added the related Extended Data Fig. 9.

Comments:

1. The diagram in Fig. 1b is misleading. It gives the impression that RES is removed from the spliceosome during the transition from Bact to BAQR, yet RES is only removed away from the pre-mRNA without being released from the spliceosome.

The RES complex loses contact with the RNA, and nearly all interfaces with the spliceosome (PRP8, and the SF3B complex), while remaining flexibly anchored via a short region of BUD13. We would refer to this event as a destabilization rather than dissociation. It is common in human spliceosomes that proteins dissociate gradually, remaining attached flexibly after fulfilling the specific function. For example, the PRP2 helicase and, to a reasonable extent, SF3A/B proteins are still present in the composition of C complexes (presumably by flexible anchoring), although they have been structurally "undocked" from the spliceosome (see, for example, the proteomic analyses from Agafonov et al., Mol Cell Biol, 2011). Therefore, we suggest referring to *destabilization* events rather than *dissociation*, whose meaning is more stringent. To avoid confusion, we explained in the legend of Fig. 1b that we indicate the recruited or destabilized subunits and that the destabilization does not equate to complete dissociation.

2. Representations of pink, green, and broken lines should be indicated in Fig. 3a.

Pink, green, and broken lines represent the PPT, PPT region bound by SF3B1 within B^{act}, and PPT equivalent region visible within yeast B^{act} structure. We have added this description in the legend of Figure 3a.

3. P.2, line 39: It is not clear to me how the authors determined that PRP2 translocates 18 nucleotides from the periphery towards the spliceosome core. Is the region of the pre-mRNA that PRP2 binds to on the Bact structure clearly visualized so that the distance of the translocation can be measured?

In the revised manuscript, we explain with more clarity our analysis. We included the Extended Data Fig. 5a-e, with a detailed description of the modeling. We also corrected an error – PRP2 has translocated 19 nucleotides, not 18, as we stated before. We apologize for the error.

In short, the pre-mRNA region that Prp2p binds is visible in the yeast B^{act} spliceosomes from Shi lab (nts 28 to 34 , PDB 7DCO, Bai et al., 2021) and consistent with the biochemistry from Soo Chen Cheng lab (Liu et al., 2012). The local resolution around PRP2 is low in Hs B^{act}, yet good enough to enable unambiguous localization of PRP2 in a virtually equivalent position as in Sc B^{act} (Zhang et al., 2018). The intron is visible until the nucleotide 19 in Hs B^{act}. Superposition between the human and yeast B^{act} complexes allows counting the missing nucleotides between 19 and PRP2's binding sites. In this way, we estimate that human PRP2 binds the nucleotides 26 – 32, *versus* 28 – 34 in yeast (there is a difference of two nucleotides between human and yeast, as the latter has a two-base insertion). Of note, the translocation of 19 nucleotides by PRP2 is consistent with the stringent requirement for at least 32-34 nucleotides downstream of the BS-A for efficient branching (Bessonov et al., RNA, 2010).

4. P.3, line 53: The description of "The first reaction.....within the C complex spliceosome" is imprecise. The C complex is the product of the first reaction, and is a stable form of the complex that can be isolated and analyzed.

We agree and have corrected the error by stating "The branching reaction occurs..... during the C complex formation" (P3, lines 50-52)".

5. P.3, line 68: The authors appear unaware of and have failed to cite previous studies on yeast related to how PRP2 may drive catalytic activation (Liu, 2012). Two other DEAH RNA helicases, PRP16 and PRP22, have also been characterized in yeast, and a similar mechanism for remodeling of the spliceosome in the second catalytic step and for the release of mature mRNA has been proposed (Chung, 2019; Schwer, 2008). The authors should discuss if this mechanism is general for DEAH RNA helicases.

We apologize and have amended the manuscript accordingly. We acknowledge the importance of this work (Liu 2012) and other papers dedicated to the Prp2p investigation in budding yeast, and we have included a thorough analysis of the catalytic activation in budding yeast (see the new section from P13). We have also depicted a diagram of the catalytic activation in yeast (Fig. 5c), by integrating previously published data and the current work.

In principle, the DEAH helicases PRP16 and PRP22 might follow a similar mechanism as PRP2, rather than acting as winches (which would agree with the earlier models derived from yeast splicing). In particular, cryo-EM structures of C complexes show that PRP16 interacts with its targets CCDC49

and CCDC94 (Cwc25p and Yju2p in yeast), at the 3' end of the branched intron, before their release. Reminiscent of how PRP2 strips away the RES complex, PRP16 might translocate in the 3'-to-5' direction to displace the two-step I factors, promoting substrate repositioning for exons ligation. Furthermore, a comparison between variants of C* complexes shows that PRP22, slightly but significantly, changes its position from the periphery of the spliceosome (Zhan, Mol Cell, 2022). Although the RNA strand bound by PRP22 is not visible due to the local low resolution, the relocation of PRP22 might suggest translocation along the intron. We discuss these aspects in the revised manuscript (P11, lines 260-272).

6. P.5, line 120: How can one explain the observation that PRP2 on Sc B^{act} exhibits the conformation corresponding to a post-ATP hydrolytic state prior to PRP2 activity?

To stall a B^{act} complex, the authors (Bai et al., Science 2020) have used recombinant Prp2p carrying the ATPase-defective mutant K252A. Because this mutation in the Walker motif prevents ATP binding, the two RecA domains are relaxed in the opened conformation, equivalent to the post-ATP hydrolytic state. We refer to the above on P5, lines 109-111.

7. P.10, line 238: PRP2....translocated to "an upstream" position.

We have corrected this error accordingly (P9, line 209)

8. P.12: "SPP2" instead of "Spp2" for consistency.

In the revised manuscript, we capitalize the names of human proteins (by following the nomenclature suggested in Kastner et al., CSHP in Biology, 2019). For instance, we refer to the human PRP2 and yeast Prp2p; or human GPKOW and yeast Spp2p. Besides, we introduce the names of the proteins by explicitly stating which is human and which is yeast, to avoid confusion.

9. P.12: This result is consistent with a previous report that the NTD of Sc PRP2 is dispensable (Edwalds-Gilbert, 2004), which should be included in the Discussion.

Interestingly, although PRP2^{NTD} is dispensable in budding yeast, Prp2p contains the "hook" element of the NTD, whose human homolog interacts with CWC22^{HEAT10} and the RES complex in B^{AQR}. As the yeast Cwc22p^{HEAT10} is required for the productive role of Prp2p (Yeh et al., Mol Cell Biol, 2011), Prp2p's hook element might play a conserved role in linking the function of Cwc22p and Prp2p in catalytic activation in yeast. The dispensable quality of NTD in yeast (Edwalds-Gilbert, 2004) may

suggest that other subunits might compensate for its function without altering the function of Prp2p. We found these correlations of potential interest and have included them in the manuscript (P14, lines 333-338).

10. The model in Fig. 5b should be explained in more detail, in particular, how the hypothetical intermediate is generated. It would also be more appropriate to place the summary of the respective results in the legend of Fig. 5b than in the Discussion.

For clarity's sake, we expanded the schematic with an additional intermediate and have explained in the legend of Fig. 5b how the intermediates were generated.

The hypothetical intermediate between B^{act} and B^{AQR} considers (i) that the molecular brake can form only after PPIL4's binding to the RNA (which in turn occurs only after dissociation of the RES complex by the translocation of PRP2) and (ii) the advance of PRP2 should progressively strip the intron from SF3B1 and promote the opening of SF3B1 towards the "loose" conformation.

We generated the hypothetical intermediate between B^{AQR} and B* by considering that Aquarius should induce complete displacement of the branch duplex from SF3B, which assumes the release of BS-A from the binding pocket on SF3B1 and SF3B1's transition from the loose to the open conformation. This transition would involve the interface between SF3B1 and PRP8, resulting in the dissociation of SF3B (together with SF3A, PPIL4, and PRP2) from PRP8. All these explanations are in the legend of Fig. 5b.

Referee #2:

This manuscript provides critical new concepts to our understanding of spliceosomal activation, i.e., the process by which, just prior to catalytic step one, the first-step nucleophile — the branch site adenosine complexed with U2 snRNA — is released from SF3b1 and moves 50A into the active site. The authors present a ~2.9Å cryo-EM structure of a human spliceosomal complex stalled by a mutation in the ATPase Aquarius. This represents a previously unknown intermediate in human spliceosome assembly/activation, which is not present in the yeast system (which does not have an Aquarius ortholog). The manuscript provides key new insights into the function of both PRP2 and Aquarius ATPases in spliceosome activation.

Lines 130-131, "Whether DEAH-box helicases, including PRP2, pull their substrates from a distance or by stepwise translocation along the RNA strand remained unclear.": This is a big point, and I want to reiterate it and say that I entirely agree with the authors' assertion here. I don't think that we have known what any spliceosomal ATPase really does, and this is a big step forward in understanding

PRP2 function. By stalling the complex at the AQR step, the authors can, for the first time, infer exactly what PRP2 translocates on: "PRP2 advanced 18 nts [on the pre-mRNA] towards the spliceosome core during the Bact-to-BAQR transition". I think that this point and its concomitant effects on the structure are incredibly important and an appropriate topic for a paper in Nature. Overall, this is one of the few spliceosome structural papers in which we actually learn new, unanticipated features of the mechanism.

Specific Comments:

1. Line 83, "Following up on our previous work 6": This needs a bit more explanation.

We are more specific in the revised manuscript by stating, "After identifying the IBC as a complex that delivers Aquarius to the spliceosome (De et al., 2015).. (P4, lines 75-77).

2. The mechanism of translocation termination of PRP2 by "molecular brake elements" proposed by the authors is very interesting. It would be useful to (briefly) compare this to what is known about the termination of translocation of other helicases. For example, is it similar, or not, to the mechanism by which MAGOH-Y14 inhibits eIF4AIII and locks it onto RNA within the exon junction complex (Ballot et al., NSMB, 2005; PMID 16170325)?

There are notable differences between the inhibition of eIF4AIII and PRP2. We mention this aspect in the revised manuscript (P10, lines 250-252).

MAGOH-Y14 partially embraces the helicase and stabilizes the closed conformation. In the case of PRP2, SKIP intercalates between the RecA domains, stabilizing the open conformation and preventing translocation and ATP hydrolysis. Besides, PPIL4 forms a brake shoe that prevents translocation by PRP2. Most likely, the two manners of inhibition respond to different needs – in one case, a DEAD box that acts primarily as an immobile clamp; in the second case, a translocase that actively walks along the RNA and should stop only after fulfilling its role.

3. The action of the ATPases as molecular clamps seems a lot like what was proposed by Melissa Moore for DExH/D proteins (Shibuya et al., NSMB, 2004; PMID 15034551).

Indeed, eIF4III appears to act as an RNA clamp and 'place holder' for the attachment of additional proteins to RNA in a sequence-independent manner. PRP2 is arrested to prevent the continuation of translocation that might dissociate or alter the branch duplex.

4. Line 377, Fig. 1, "Key subunits are color-coded": This could be clarified. I understand that the colors in the molecule structure schematic are a rough match with the colors for the components list, shown in a, lower. However, there are some inconsistencies. For example, PRP2 is shown in brown, but it is not color-matched in Fig. 1b, where PRP2 is inconsistently written in red. Why is the RES complex indicated in light blue, when the only protein component included, BUD13, is dark blue? Overall, there are too many colors and shades of colors here to be able to follow.

We fully agree and have amended Fig.1 by removing inconsistencies in color coding. We believe that the revised Fig. 1a is much easier to follow.

5. Lines 268-271: There is a high richness of new detail provided throughout. However, one statement, "The two helicases must coordinate to liberate the duplex from SF3B-RES constraints and hand it over to the catalytic center, while keeping it unaltered – a task that is best accomplished sequentially, by two molecular motors", leaves me wanting more info. Why is this "best accomplished" by two molecular motors? This is a critical difference between mammals and yeast, and I think it deserves to be explained further. Here, this is stated as a fact; as such, it needs further explanation. Alternatively, this could be stated as a conjecture that is, so far, without adequate explanation.

We fully agree and have updated the manuscript, discussing thoroughly why human catalytic activation requires two helicases, and the putative mechanism of Aquarius (P12-P13, lines 273-315). Furthermore, we dedicated a section comparing the yeast and human catalytic activation (P13-P14, lines 317 – 345). We have also added a diagram about the yeast catalytic activation (Fig. 5c) that can be readily compared with the human system (Fig. 5b)

Line 378: "spliceosomes" should be "spliceosome".

We have corrected the error accordingly (P19, lines 518-519).

Referee #3:

The manuscript by Schmitzova et al. presents the structure of a new spliceosome assembly intermediate, Baqr, captured via a dominant-negative mutation in the helicase Aquarius. Aquarius is the only SF1-family helicase that is involved in splicing and its function is not well understood, in particular, that it is absent in some model organisms (e.g., *S. cerevisiae*). The complex presented by the authors was stalled between Bact and B*/C states and provides important mechanistic insights into this transition.

Some of the key findings include:

1. Dissection of the Bact to B* transition into two steps, implying that PRP2 and Aquarius act in a sequential manner.
2. PRP2 was captured in a position suggesting that it translocates along the pre-mRNA in order to displace a specific subset of splicing, including the RES complex. It was previously proposed that PRP2 would remain attached to SF3B1 and act on the branch helix by pulling from a distance, but the exact mechanism of this remodeling remained elusive. The Baqr structure provides a conceptually new mechanism for this rearrangement.
3. PRP2 action alone is sufficient to unfasten the branch helix from the SF3b complex by re-opening SF3B1 and therefore priming it for further remodeling.
4. SKIP and PPIL4 associate with intron RNA exiting PRP2 and could act as a molecular brake to achieve precise control over the extent of PRP2 translocation.

Overall it is a beautiful work providing new and exciting insights into the mechanism of one of the key steps in spliceosome assembly. Technically, the biochemical and structural aspects of the work are of high quality. The manuscript is clearly written and easy to follow, with all findings accompanied by appropriate figures.

I would recommend publication of this work in Nature upon addressing some of my comments below. Major points:

1. While I understand the authors' logic that PRP2 is required for the Bact to Baqr transition and that Aquarius is needed further down the line, I am not entirely convinced if one can simplify the whole process by stating that the two helicases act in a sequential manner. One could imagine a situation

where the action of Aquarius could only be needed to enable PRP2 to continue its translocation until the branch helix is completely liberated. This could happen, for example, by translocating the remote parts of the intron upstream from the branch site to release some potential tensions in the system to allow PRP2 to continue its unwinding. In other words, it is possible that IBC-K829A mutation would keep IBC locked on one position of the intron, preventing PRP2 from further translocation. What would speak in favor of such a hypothesis is the fact that Prp2 alone is capable of performing catalytic activation in some species (i.e., *S. cerevisiae*) without the need for Aquarius. The sequential action of the two helicases is an important point of the paper, and alternative scenarios should be at least thoroughly considered in the Discussion. Ideally, the authors should try to come up with a feasible experiment showing that PRP2 is not needed to complete the catalytic activation after initial translocation or prove that Aquarius is really needed to complete it.

The knowledge that PRP2's ortholog in budding yeast (Prp2p) promotes catalytic activation without needing a second helicase was instrumental for interpreting the role of Aquarius in human splicing. In the revised manuscript, we thoroughly discuss the parallels between yeast and human (also asked by referee 1; P13-P14, lines 319-345).

We suggest that Prp2p suffices for catalytic activation because the constituents of the molecular brake do not exist in budding yeast (i.e., were lost during the evolution). Consequently, there is no need for a second helicase, such as Aquarius, to unlock the stalled spliceosome. Without a brake, Prp2p can continue translocation until the complete liberation of the branch duplex in budding yeast. We also include the structure-based model of the catalytic activation in budding yeast, derived by integrating biochemical and genetic, and the structure of B^{AQR} (Fig. 5c).

One could imagine a situation where the action of Aquarius could only be needed to enable PRP2 to continue its translocation until the branch helix is completely liberated. This could happen, for example, by translocating the remote parts of the intron upstream from the branch site to release some potential tensions in the system to allow PRP2 to continue its unwinding.

We cannot exclude a certain chronological overlap in the action of the two helicases. The structure of B^{AQR} complex suggests that propagation of Aquarius's influence downstream the BS-A, where PRP2 resides, is nevertheless difficult. Because the branch duplex is still anchored to SF3B, with the BS-A bound to its pocket from SF3B1, any possible influence of Aquarius on PRP2 should first affect (possibly detach) the branch duplex, before reaching the downstream PRP2 (see the lines of structural communication from Extended Data Fig. 9a). We would instead suggest that by changing the tension in the intron, Aquarius might produce an additional loosening of the SF3B1:branch duplex interaction, facilitating the release of the duplex by the concomitant PRP2's translocation. In the revised manuscript, we discuss how Aquarius might promote the release of the branch helix (see the revised

section from P12 and P13) and have explicitly stated the possibility of overlap between the activities of the two helicases (P13, lines 307-310).

Ideally, the authors should try to come up with a feasible experiment showing that PRP2 is not needed to complete the catalytic activation after initial translocation or prove that Aquarius is really needed to complete it.

We have conceived a biochemical experiment that, in principle, might differentiate between the actions of the two helicases. In short, we attempted to chase the two helicases in splicing extracts by considering the distinct hydrolytic preferences of PRP2 and Aquarius for different nucleoside triphosphates. While PRP2 can hydrolyse both ATP and GTP, Aquarius hydrolyses only ATP. To start, we attempted to purify a spliceosome intermediate that precedes PRP2 but does not involve any additional helicase (e.g., BRR2) for progressing in the splicing pathway. This condition is met by a recently reported pre-B^{act} complex that stalled reversibly with a phosphatase inhibitor (NSC95397; Townsend et al., Science, 2020). Adding a micrococcal nuclease-treated HeLa nuclear extract (which contains a mixture of RNA-free splicing factors) to the inhibited pre-B^{act} complex might, in principle, restore splicing only in the presence of ATP (and not in the presence of GTP). We have performed the experiment several times, but the overall signal was far too weak for a reliable detection (for all samples, including the various controls), possibly because the splicing extract is suboptimal or the scale was too small, or a combination of these factors. In our experience, chasing human spliceosomes can be a very tedious and time-consuming task that requires a dedicated project. While we agree that such an experiment would be ideal, it seems not feasible for the purpose of this manuscript.

2. The authors propose that SKIP and PPIL4 act as a molecular brake that stops PRP2 from unwinding branch duplex and suggest that both proteins act as negative regulators of PRP2 in a context-dependent manner. This is potentially a very important point as such a regulation mechanism is conceptually new and might also be used by other spliceosomal helicases. However, based on the structure alone, the causality remains ambiguous here (i.e., it is not clear to me if SKIP/PPIL4 could bind to PRP2 and actively induce its opening/inhibition, or do they just associate with the repositioned PRP2, which is inhibited from translocating further by some other means). It would significantly strengthen the manuscript if the authors provided some experimental evidence that SKIP and PPIL4 can indeed influence the intrinsic activity of PRP2 (for example, by an *in vitro* helicase assay).

To test whether the recombinant PPIL4 and SKIP form a molecular brake outside spliceosomes, thereby inactivating PRP2's unwinding, we prepared constructs for recombinant expression in insect cells of PRP2, PPIL4, SKIP, and GPKOW (the G-patch cofactor of PRP2). We succeeded in expressing and purifying to homogeneity PRP2 (137-1022, corresponding to the region resolved in

the cryo-EM map), GPKOW and PPIL4. Full-length PRP2 is expressed in very low amounts and virtually impossible to purify, while SKIP (predicted to be largely disordered in isolation) proved unstable and prone to massive degradation (detected as small amounts of fragments associated with chaperones).

We used in our assays PRP2 (137-1022), after ensuring that the protein was free from potential nucleic acid contaminants. In agreement with the B^{AQR} structure, we show that PRP2 and PPIL4 form a complex. We evidenced the complex both by co-expression, and by mixing the two proteins, followed by size exclusion chromatography (Extended Data Figure 7a-c; the complete gels are in the Supplementary Fig. 2). We can also evidence for the first time that PRP2 and GPKOW form a stable complex, consistent with previous biochemical observation on the yeast orthologs Prp2p and Spp2p (Extended Data Figure 7f). The above complexes and the individual proteins can shift RNA by EMSA (Extended Data Figure 7d, e, g). Taken together, these experiments indicate that PRP2 is expressed as a well-folded protein that interacts with predicted ligands.

Next, we assessed the unwinding activity of PRP2. Not entirely unexpected, human PRP2 shows no unwinding activity in isolation, neither on its own nor in complex with GPKOW (Extended Data Figure 7h-l). While attempts of testing activity of human PRP2 were not reported before, two labs have reported that the yeast recombinant Prp2p orthologue has no helicase activity in isolation (Kim et al., EMBO J, 1992; Warkocki et al., Genes&Dev, 2015). PRP2 and Prp2p manifests translocation only in the presence of spliceosomes for reasons not understood so far. Until now, the B^{AQR} spliceosomes provided the only direct evidence that PRP2 can indeed translocate.

3. The key question that remains open is what is the role of Aquarius in the catalytic activation process? Although Aquarius was used by the authors to stall spliceosomes, the manuscript does not discuss its role in this process as all the analyses are focussed on PRP2. What is particularly puzzling is the fact that Aquarius binds upstream of the branch site, and given that it has 3'-5' RNA unwinding activity (De et al., NSMB, 2015), it is very difficult to imagine how it could exert any force on the branch helix, which still needs to be remodeled. By translocating with this polarity, it should create slack rather than tension between itself and the branch helix (see also point 1). I believe that the authors should comment on this aspect of their work. In particular, since the title "Structural basis of catalytic activation in human splicing" implies insights into the whole process of catalytic activation, but here only part of this process was analyzed.

We fully agree and have updated the manuscript accordingly. From a mechanistic perspective, the most straightforward explanation is that Aquarius pulls the branch duplex remotely, by exerting a

wincing activity from its upstream location on the intron, at the B^{AQR} to B*/C transition (Extended Data Fig. 9a c). Once the BS-A is extracted from its pocket on SF3B1, the HEAT domain could spontaneously shift from the "loose" to the "open" conformation (the latter is energetically more favorable). As the open conformation is not compatible with PRP8 binding (due to rearrangements of HEAT repeats), BS-A dissociation would likely induce further dissociation of SF3B1, along with the entire SF3A/B (Extended Data Fig. 9a-c and Extended Data Fig. 6d). This scenario implies that Aquarius translocate with 5'-to-3' polarity in the spliceosome, similar to the phylogenetically related helicase Upf1.

The 5'-to-3' translocation is not mutually exclusive with the apparent 3'-to-5' polarity exhibited by the recombinant Aquarius in unwinding assays (De et al., 2015). It has been previously noticed that the mutation Y1196A abolishes this unwinding activity while not affecting splicing, indicating that the significance of the unwinding assay for splicing remains unclear (De et al., 2015). Furthermore, it is not unprecedented that the relevance of unwinding assays on physiological translocation is limited. For instance, the splicing DEAH helicase PRP43 requires either 5' or 3' overhangs to unwind duplexes in isolation, exhibiting an apparent double polarity of translocation (Tanaka and Schwer, *Biochemistry*, 2006). However, PRP43 acts exclusively with 3'-to-5' polarity once incorporated in the spliceosomes. Furthermore, in isolation, PRP2 has no unwinding activity, although it translocates with 3'-to-5' polarity on spliceosomes (Kim et al., *EMBO J*, 1992; Warkocki et al., *Genes&Dev*, 2015; and this work). We, therefore, find plausible the possibility of 5'-to-3' translocation of Aquarius during the catalytic activation.

An alternative scenario is that Aquarius may use 3'-5' translocation in the spliceosome, and Y1196 is important only for duplex separation but dispensable for unidirectional translocation on single-stranded RNA. Although this polarity would produce slack instead of tension on the intron, the changes in the position and orientation of Aquarius, likely inherent to the motor activity, might affect interfaces between Aquarius and other components (especially SF3A and SF3B components), thus propagating a change all the way to the SF3B1 (like a "domino effect"; Extended Data Figure 9a), perturbing the contacts between SF3B1 and PRP8, and ultimately promoting dissociation of the SF3A/B complexes. In both scenarios presented above, SF3B1 would likely undergo the "loose" to "open" transition upon dissociating from the spliceosome.

We discuss the mechanism of Aquarius in the revised manuscript (P12, P13), including the implications on our understanding of SF3B1's cyclic pathway of transitions in splicing (P15).

Minor comments:

1. Line 68, "by a mechanism that remains largely hypothetical": I would suggest rephrasing to "by a mechanism that remains largely unknown".

We agree and have rephrased as suggested (P3, lines 60-62)

2. Line 167, "Surprisingly, after releasing the RES complex from RNA and former protein contacts, the RES complex binds, via BUD13531-534, to a composite structure formed by the hook of PRP2NTD, the slightly relocated C-terminal MA3 domain of CWC22 and PRP8 (Fig. 3a and Extended Data Fig. 13b,c)": This is a bit misleading in suggesting that the entire RES complex is relocated, while it is only a small fragment of Bud13 that rebinds. This should be rephrased to avoid any confusion.

To avoid confusion, we revised the text as follows:

"Surprisingly, after destabilizing the RES complex from RNA and former protein contacts, only a short stretch of BUD13 is visible (residues 530-557) in B^{AQR}, bound to a composite structure formed by the hook of PRP2^{NTD}, PRP8 and the slightly relocated MA3 domain of CWC22 (Fig. 3a and Extended Data Fig. 4e,f). However, proteomic analysis shows that the entire RES complex remains bound to the spliceosome (Supplementary Data S1). Thus, BUD13⁵³⁰⁻⁵⁵⁷ anchors flexibly the RES complex to the spliceosome, conditioned by the PRP2 hook's binding to PRP8, whose presence may serve as a signature for the occurrence of PRP2's translocation." (P7, lines 151-157)

3. It would be helpful to see some examples of high-resolution fitting, as current cryo-EM density snapshots (Ext. Data Fig. 8) show mostly secondary structure elements and do not support high-resolution data claimed by the FSC. This is relevant given the highly anisotropic angular distribution of the 3D reconstruction, which could cause a mismatch between the resolution estimate and observed features.

We have added examples of high-resolution fitting, where side chains or the small molecule IP6 (commonly found bound to PRP8) are clearly fitted in the density (Extended Data Figure 3d).

4. There is a numbering problem with Ext. Data Figs. (12, 14, 14); 13 is missing.

In the revised manuscript, we have reorganized the Extended Data figures to avoid exceeding nine figures plus one table (as required by the journal).

5. In Table S2 some entries are colour-coded, but the meaning of that is not explained.

The entries colored in purple represent subunits that were modeled only in the overall maps, not in the high-resolution core map. The former Table S2 has become Data S2 (spreadsheet), and the caption is included in the Supplementary Information file.

Reviewer Reports on the First Revision:

Referees' comments:

Referee #1 (Remarks to the Author):

In the revised manuscript, the authors extend the Discussion in a great deal, and provide thorough and insightful views on the mechanism of catalytic activation of the spliceosome in both human and yeast. The authors have adequately addressed all my concerns. The work is well done, and the manuscript is much improved. I have no further comments on the manuscript.

Referee #2 (Remarks to the Author):

I really liked this manuscript the first time, and I still think that it provides critical new concepts to our understanding of spliceosomal activation. I find the authors' answers to the reviewers' comments, and the additional discussion points, thoughtful and complete, and they addressed all of my concerns. I think that the new and unanticipated mechanistic features of the spliceosome function presented here will be appreciated by a broad audience.

Referee #3 (Remarks to the Author):

The authors have been very responsive to my comments and concerns.

In particular, the inclusion of a detailed discussion of the possible role of Aquarius in catalytic activation and its translocation polarity greatly strengthens the manuscript and gives the reader a better mechanistic insight into this process.

Additional experiments were performed to confirm the hypothesis that PPIL4, SKIP, and PRPF2 NTD act as a molecular brake for the helicase translocation. The authors were unable to detect PRPF2 unwinding activity *in vitro*, and consequently, their attempts to address this issue failed. Nevertheless, these experiments were carefully designed and executed and provide valuable additional data that may be useful to the community. I believe that proving the proposed mechanism experimentally could be a cumbersome task, which goes beyond the scope of this manuscript.

Overall, the revised manuscript is much better than the previous version, and I would like to recommend it for publication in Nature. These are very exciting findings for the splicing community.